# Variational Hetero-Encoder Randomized GANs for Joint Image-Text Modeling

**Hao Zhang, Bo Chen,**[*] **Long Tian, Zhengjue Wang**
National Laboratory of Radar Signal Processing
Xidian University, Xian, China
`zhanghao_xidian@163.com`  `bchen@mail.xidian.edu.cn`
`tianlong_xidian@163.com`  `zhengjuewang@163.com`

**Mingyuan Zhou**
McCombs School of Business
The University of Texas at Austin, Austin, TX 78712, USA
`mingyuan.zhou@mccombs.utexas.edu`

## Abstract

For bidirectional joint image-text modeling, we develop variational hetero-encoder (VHE) randomized generative adversarial network (GAN), a versatile deep generative model that integrates a probabilistic text decoder, probabilistic image encoder, and GAN into a coherent end-to-end multi-modality learning framework. VHE randomized GAN (VHE-GAN) encodes an image to decode its associated text, and feeds the variational posterior as the source of randomness into the GAN image generator. We plug three off-the-shelf modules, including a deep topic model, a ladder-structured image encoder, and StackGAN++, into VHE-GAN, which already achieves competitive performance. This further motivates the development of VHE-raster-scan-GAN that generates photo-realistic images in not only a multi-scale low-to-high-resolution manner, but also a hierarchical-semantic coarse-to-fine fashion. By capturing and relating hierarchical semantic and visual concepts with end-to-end training, VHE-raster-scan-GAN achieves state-of-the-art performance in a wide variety of image-text multi-modality learning and generation tasks.

## 1 Introduction

Images and texts commonly occur together in the real world. There exists a wide variety of deep neural network based unidirectional methods that model images (texts) given texts (images) (Gomez et al., 2017; Kiros & Szepesvari, 2012; Reed et al., 2016; Xu et al., 2018; Zhang et al., 2017a). There also exist probabilistic graphic model based bidirectional methods (Srivastava & Salakhutdinov, 2012b;a; Wang et al., 2018) that capture the joint distribution of images and texts. These bidirectional methods, however, often make restrictive parametric assumptions that limit their image generation ability. Exploiting recent progress on deep probabilistic models and variational inference (Kingma & Welling, 2014; Zhou et al., 2016; Zhang et al., 2018a; Goodfellow et al., 2014; Zhang et al., 2017b), we propose an end-to-end learning framework to construct multi-modality deep generative models that can not only generate vivid image-text pairs, but also achieve state-of-the-art results on various unidirectional tasks (Srivastava & Salakhutdinov, 2012b;a; Wang et al., 2018; Gomez et al., 2017; Xu et al., 2018; Zhang et al., 2017a;b; Verma et al., 2018; Zhang et al., 2018b), such as generating photo-realistic images given texts and performing text-based zero-shot learning.

To extract and relate semantic and visual concepts, we first introduce variational hetero-encoder (VHE) that encodes an image to decode its textual description (*e.g.*, tags, sentences, binary attributes, and long documents), where the probabilistic encoder and decoder are jointly optimized using variational inference (Blei et al., 2017; Hoffman et al., 2013; Kingma & Welling, 2014; Rezende et al., 2014). The latent representation of VHE can be sampled from either the variational posterior provided

---

[*]Corresponding author

by the image encoder given an image input, or the posterior of the text decoder via MCMC given a text input. VHE by construction has the ability to generate texts given images. To further enhance its text generation performance and allow synthesizing photo-realistic images given an image, text, or random noise, we feed the variational posterior of VHE in lieu of random noise as the source of randomness into the image generator of a generative adversarial network (GAN) (Goodfellow et al., 2014). We refer to this new modeling framework as VHE randomized GAN (VHE-GAN).

Off-the-shelf text decoders, image encoders, and GANs can be directly plugged into the VHE-GAN framework for end-to-end multi-modality learning. To begin with, as shown in Figs. 1(a) and 1(b), we construct VHE-StackGAN++ by using the Poisson gamma belief network (PGBN) (Zhou et al., 2016) as the VHE text decoder, using the Weibull upward-downward variational encoder (Zhang et al., 2018a) as the VHE image encoder, and feeding the concatenation of the multi-stochastic-layer latent representation of the VHE as the source of randomness into the image generator of StackGAN++ (Zhang et al., 2017b). While VHE-StackGAN++ already achieves very attractive performance, we find that its performance can be clearly boosted by better exploiting the multi-stochastic-layer semantically meaningful hierarchical latent structure of the PGBN text decoder. To this end, as shown in Figs. 1(a) and 1(c), we develop VHE-raster-scan-GAN to perform image generation in not only a multi-scale low-to-high-resolution manner in each layer, as done by StackGAN++, but also a hierarchical-semantic coarse-to-fine fashion across layers, a unique feature distinguishing it from existing methods. Consequently, not only can VHE-raster-scan-GAN generate vivid high-resolution images with better details, but also build interpretable hierarchical semantic-visual relationships between the generated images and texts.

Our main contributions include: 1) VHE-GAN that provides a plug-and-play framework to integrate off-the-shelf probabilistic decoders, variational encoders, and GANs for end-to-end bidirectional multi-modality learning; the shared latent space can be inferred either by image encoder $q(\boldsymbol{z} \mid \boldsymbol{x})$, if given images, or by Gibbs sampling from the conditional posterior of text decoder $p(\boldsymbol{t} \mid \boldsymbol{z})$, if given texts; 2) VHE-raster-scan-GAN that captures and relates hierarchical semantic and visual concepts to achieve state-of-the-art results in various unidirectional and bidirectional image-text modeling tasks.

## 2 VARIATIONAL HETERO-ENCODER RANDOMIZED GANS

VAEs and GANs are two distinct types of deep generative models. Consisting of a generator (decoder) $p(\boldsymbol{x} \mid \boldsymbol{z})$, a prior $p(\boldsymbol{z})$, and an inference network (encoder) $q(\boldsymbol{z} \mid \boldsymbol{x})$ that is used to approximate the posterior $p(\boldsymbol{z} \mid \boldsymbol{x})$, VAEs (Kingma & Welling, 2014; Rezende et al., 2014) are optimized by maximizing the evidence lower bound (ELBO) as

$$\text{ELBO} = \mathbb{E}_{\boldsymbol{x} \sim p_{\text{data}}(\boldsymbol{x})}[\mathcal{L}(\boldsymbol{x})], \quad \mathcal{L}(\boldsymbol{x}) := \mathbb{E}_{\boldsymbol{z} \sim q(\boldsymbol{z} \mid \boldsymbol{x})}\left[\ln p(\boldsymbol{x} \mid \boldsymbol{z})\right] - \text{KL}\left[q(\boldsymbol{z} \mid \boldsymbol{x})||p(\boldsymbol{z})\right], \quad (1)$$

where $p_{\text{data}}(\boldsymbol{x}) = \sum_{i=1}^{N} \frac{1}{N}\delta_{\boldsymbol{x}_i}$ represents the empirical data distribution. Distinct from VAEs that make parametric assumptions on data distribution and perform posterior inference, GANs in general use implicit data distribution and do not provide meaningful latent representations (Goodfellow et al., 2014); they learn both a generator $G$ and a discriminator $D$ by optimizing a mini-max objective as

$$\min_G \max_D \{\mathbb{E}_{\boldsymbol{x} \sim p_{\text{data}}(\boldsymbol{x})}\left[\ln D(\boldsymbol{x})\right] + \mathbb{E}_{\boldsymbol{z} \sim p(\boldsymbol{z})}\left[\ln(1 - D(G(\boldsymbol{z})))\right]\}, \quad (2)$$

where $p(\boldsymbol{z})$ is a random noise distribution that acts as the source of randomness for data generation.

### 2.1 VHE-GAN OBJECTIVE FUNCTION FOR END-TO-END MULTI-MODALITY LEARNING

Below we show how to construct VHE-GAN to jointly model images $\boldsymbol{x}$ and their associated texts $\boldsymbol{t}$, capturing and relating hierarchical semantic and visual concepts. First, we modify the usual VAE into VHE, optimizing a lower bound of the text log-marginal-likelihood $\mathbb{E}_{\boldsymbol{t} \sim p_{\text{data}}(\boldsymbol{t})}\left[\ln p(\boldsymbol{t})\right]$ as

$$\text{ELBO}_{\text{vhe}} = \mathbb{E}_{p_{\text{data}}(\boldsymbol{t},\boldsymbol{x})}[\mathcal{L}_{\text{vhe}}(\boldsymbol{t}, \boldsymbol{x})], \quad \mathcal{L}_{\text{vhe}}(\boldsymbol{t}, \boldsymbol{x}) := \mathbb{E}_{\boldsymbol{z} \sim q(\boldsymbol{z} \mid \boldsymbol{x})}\left[\ln p(\boldsymbol{t} \mid \boldsymbol{z})\right] - \text{KL}\left[q(\boldsymbol{z} \mid \boldsymbol{x})||p(\boldsymbol{z})\right], (3)$$

where $p(\boldsymbol{t} \mid \boldsymbol{z})$ is the text decoder, $p(\boldsymbol{z})$ is the prior, $p(\boldsymbol{t}) = \mathbb{E}_{\boldsymbol{z} \sim p(\boldsymbol{z})}[p(\boldsymbol{t} \mid \boldsymbol{z})]$, and $\mathcal{L}_{\text{vhe}}(\boldsymbol{t}, \boldsymbol{x}) \leq \ln \mathbb{E}_{\boldsymbol{z} \sim q(\boldsymbol{z} \mid \boldsymbol{x})}\left[\frac{p(\boldsymbol{t} \mid \boldsymbol{z})p(\boldsymbol{z})}{q(\boldsymbol{z} \mid \boldsymbol{x})}\right] = \ln p(\boldsymbol{t})$. Second, the image encoder $q(\boldsymbol{z} \mid \boldsymbol{x})$, which encodes image $\boldsymbol{x}$ into its latent representation $\boldsymbol{z}$, is used to approximate the posterior $p(\boldsymbol{z} \mid \boldsymbol{t}) = p(\boldsymbol{t} \mid \boldsymbol{z})p(\boldsymbol{z})/p(\boldsymbol{t})$. Third, variational posterior $q(\boldsymbol{z} \mid \boldsymbol{x})$ in lieu of random noise $p(\boldsymbol{z})$ is fed as the source of randomness into the GAN image generator. Combing these three steps, with the parameters of the image encoder

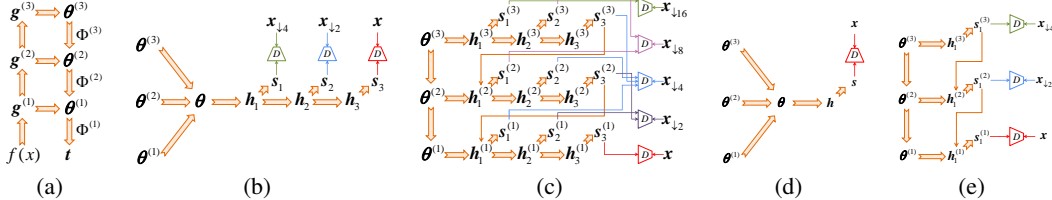

Figure 1: Illustration of (a) VHE, (b) StackGAN++, (c) raster-scan-GAN, (d) vanilla-GAN, and (e) simple-raster-scan-GAN. VHE-raster-scan-GAN consists of (a) and (c). $\boldsymbol{x}_{\downarrow d}$ is down-sampled from $\boldsymbol{x}$ with scaling factor $d$. VHE-StackGAN++, consisting of (a) and (b), VHE-vanilla-GAN, consisting of (a) and (d), and VHE-simple-raster-scan-GAN, consisting of (a) and (e), are all used for ablation studies.

$q(\boldsymbol{z}\,|\,\boldsymbol{x})$, text decoder $p(\boldsymbol{t}\,|\,\boldsymbol{z})$, and GAN generator denoted by $E$, $G_{\text{vae}}$, and $G_{\text{gan}}$, respectively, we express the objective function of VHE-GAN for joint image-text end-to-end learning as

$$\min_{E,G_{\text{vae}},G_{\text{gan}}}\ \max_{D}\mathbb{E}_{p_{\text{data}}(\boldsymbol{t},\boldsymbol{x})}[\mathcal{L}(\boldsymbol{t},\boldsymbol{x})],$$

$$\mathcal{L}(\boldsymbol{t},\boldsymbol{x}) := \ln D(\boldsymbol{x}) + \text{KL}\left[q(\boldsymbol{z}\,|\,\boldsymbol{x})||p(\boldsymbol{z})\right] + \mathbb{E}_{\boldsymbol{z}\sim q(\boldsymbol{z}\,|\,\boldsymbol{x})}\left[\ln(1 - D(G_{\text{gan}}(\boldsymbol{z}))) - \ln p(\boldsymbol{t}\,|\,\boldsymbol{z})\right]. \quad (4)$$

Note the objective function in (4) implies a data-triple-reuse training strategy, which uses the same data mini-batch in each stochastic gradient update iteration to jointly train the VHE, GAN discriminator, and GAN generator; see a related objective function, shown in (10) of Appendix A, that is resulted from naively combining the VHE and GAN training objectives. In VHE-GAN, the optimization of the encoder parameter $E$ is related to not only the VHE's ELBO, but also the GAN mini-max objective function, forcing the variational posterior $q(\boldsymbol{z}\,|\,\boldsymbol{x})$ to serve as a bridge between VHE and GAN, allowing them to help each other. Although there are some models (Mescheder et al., 2017; Makhzani et al., 2015; Tolstikhin et al., 2018; Dumoulin et al., 2017; Donahue et al., 2017; Che et al., 2017; Srivastava et al., 2017; Grover et al., 2018; Larsen et al., 2016; Huang et al., 2018) combining VAEs and GANs in various ways, they focus on single-modality tasks while the VHE-GAN on two different modalities. In Appendix A, we analyze the properties of the VHE-GAN objective function and discuss related works. Below we develop two different VHE-GANs, one integrates off-the-shelf modules, while the other introduces new interpretable hierarchical latent structure.

## 2.2 VHE-STACKGAN++ WITH OFF-THE-SHELF MODULES

As shown in Figs. 1(a) and 1(b), we first construct VHE-StackGAN++ by plugging into VHE-GAN three off-the-shelf modules, including a deep topic model (Zhou et al., 2016), a ladder-structured encoder (Zhang et al., 2018a), and StackGAN++ (Zhang et al., 2017b). For text analysis, both sequence models and topic models are widely used. Sequence models (Bengio et al., 2003) often represent each document as a sequence of word embedding vectors, capturing local dependency structures with some type of recurrent neural networks (RNNs), such as long short-term memory (LSTM) (Hochreiter & Schmidhuber, 1997). Topic models such as latent Dirichlet allocation (LDA) (Blei et al., 2003) often represent each document as a bag of words (BoW), capturing global word cooccurrence patterns into latent topics. Suitable for capturing local dependency structure, existing sequence models often have difficulty in capturing long-range word dependencies and hence macro-level information, such as global word cooccurrence patterns (*i.e.*, topics), especially for long documents. By contrast, while topic models ignore word order, they are very effective in capturing latent topics, which are often directly related to macro-level visual information (Gomez et al., 2017; Dieng et al., 2017; Lau et al., 2017). Moreover, topic models can be applied to not only sequential texts, such as few sentences (Wang et al., 2009; Jin et al., 2015) and long documents (Zhou et al., 2016), but also non-sequential ones, such as textual tags (Srivastava & Salakhutdinov, 2012a; 2014; Wang et al., 2018) and binary attributes (Elhoseiny et al., 2017b; Zhu et al., 2018). For this reason, for the VHE text decoder, we choose PGBN (Zhou et al., 2016), a state-of-the-art topic model that can also be represented as a multi-stochastic-layer deep generalization of LDA (Cong et al., 2017). We complete VHE-StackGAN++ by choosing the Weibull upward-downward variational encoder (Zhang et al., 2018a) as the VHE image encoder, and feeding the concatenation of all the hidden layers of PGBN as the source of randomness to the image generator of StackGAN++ (Zhang et al., 2017b).

As in Fig. 1, we use a VHE that encodes an image into a deterministic-upward–stochastic-downward ladder-structured latent representation, which is used to decode the corresponding text. Specifically, we represent each document as a BoW high-dimensional sparse count vector $\boldsymbol{t}_n \in \mathbb{Z}^{K_0}$, where

$\mathbb{Z} = \{0, 1, \cdots\}$ and $K_0$ is the vocabulary size. For the VHE text decoder, we choose to use PGBN to extract hierarchical latent representation from $\boldsymbol{t}_n$. PGBN consists of multiple gamma distributed stochastic hidden layers, generalizing the "shallow" Poisson factor analysis (Zhou et al., 2012; Zhou & Carin, 2015) into a deep setting. PGBN with $L$ hidden layers, from top to bottom, is expressed as

$$\boldsymbol{\theta}_n^{(L)} \sim \text{Gam}\left(\boldsymbol{r}, 1/s_n^{(L+1)}\right), \ \boldsymbol{r} \sim \text{Gam}(\gamma_0/K_L, 1/s_0),$$

$$\boldsymbol{\theta}_n^{(l)} \sim \text{Gam}\left(\boldsymbol{\Phi}^{(l+1)}\boldsymbol{\theta}_n^{(l+1)}, 1/s_n^{(l+1)}\right), l = L-1, \cdots, 2, 1, \ \ \boldsymbol{t}_n \sim \text{Pois}\left(\boldsymbol{\Phi}^{(1)}\boldsymbol{\theta}_n^{(1)}\right), \quad (5)$$

where the hidden units $\boldsymbol{\theta}_n^{(l)} \in \mathbb{R}_+^{K_l}$ of layer $l$ are factorized under the gamma likelihood into the product of topics $\boldsymbol{\Phi}^{(l)} \in \mathbb{R}_+^{K_{l-1} \times K_l}$ and hidden units of the next layer, $\mathbb{R}_+ = \{x, x \geq 0\}$, $s_n^{(l)} > 0$, and $K_l$ is the number of topics of layer $l$. If the texts are represented as binary attribute vectors $\boldsymbol{b}_n$, we can add a Bernoulli-Poisson link layer as $\boldsymbol{b}_n = \boldsymbol{1}(\boldsymbol{t}_n \geq 1)$ (Zhou, 2015; Zhou et al., 2016). We place a Dirichlet prior on each column of $\boldsymbol{\Phi}^{(l)}$. The topics can be organized into a directed acyclic graph (DAG), whose node $k$ at layer $l$ can be visualized with the top words of $\left[\prod_{t=1}^{l-1} \boldsymbol{\Phi}^{(t)}\right]\boldsymbol{\phi}_k^{(l)}$; the topics tend to be very general in the top layer and become increasingly more specific when moving downwards. This semantically meaningful latent hierarchy provides unique opportunities to build a better image generator by coupling the semantic hierarchical structures with visual ones.

Let us denote $\boldsymbol{\Phi} = \{\boldsymbol{\Phi}^{(1)}, \ldots, \boldsymbol{\Phi}^{(L)}, \boldsymbol{r}\}$ as the set of global parameters of PGBN shown in (5). Given $\boldsymbol{\Phi}$, we adopt the inference in Zhang et al. (2018a) to build an Weibull upward-downward variational image encoder as $\prod_{n=1}^N \prod_{l=1}^L q(\boldsymbol{\theta}_n^{(l)} \,|\, \boldsymbol{x}_n, \boldsymbol{\Phi}^{(l+1)}, \boldsymbol{\theta}_n^{(l+1)})$, where $\boldsymbol{\Phi}^{(L+1)} := \boldsymbol{r}$, $\boldsymbol{\theta}_n^{(L+1)} := \emptyset$, and

$$q(\boldsymbol{\theta}_n^{(l)} \,|\, \boldsymbol{x}_n, \boldsymbol{\Phi}^{(l+1)}, \boldsymbol{\theta}_n^{(l+1)}) = \text{Weibull}(\boldsymbol{k}_n^{(l)} + \boldsymbol{\Phi}^{(l+1)}\boldsymbol{\theta}_n^{(l+1)}, \boldsymbol{\lambda}_n^{(l)}). \quad (6)$$

The Weibull distribution is used to approximate the gamma distributed conditional posterior, and its parameters $\boldsymbol{k}_n^{(l)}, \boldsymbol{\lambda}_n^{(l)} \in \mathbb{R}^{K_l}$ are deterministically transformed from the convolutional neural network (CNN) image features $f(\boldsymbol{x}_n)$ (Szegedy et al., 2016), as shown in Fig. 1(a) and described in Appendix D.1. We denote $\boldsymbol{\Omega}$ as the set of encoder parameters. We refer to Zhang et al. (2018a) for more details about this deterministic-upward–stochastic-downward ladder-structured inference network, which is distinct from a usual inference network that has a pure bottom-up structure and only interacts with the generative model via the ELBO (Kingma & Welling, 2014; Gulrajani et al., 2017).

The multi-stochastic-layer latent representation $\boldsymbol{z} = \{\boldsymbol{\theta}^{(l)}\}_{l=1}^L$ is the bridge between two modalities. As shown in Fig. 1(b), VHE-StackGAN++ simply randomizes the image generator of StackGAN++ (Zhang et al., 2017b) with the concatenated vector $\boldsymbol{\theta} = \left[\boldsymbol{\theta}^{(1)}, \cdots, \boldsymbol{\theta}^{(L)}\right]$. We provide the overall objective function in (15) of Appendix D.2. Note that existing neural-network-based models (Gomez et al., 2017; Xu et al., 2018; Zhang et al., 2017a;b; Verma et al., 2018; Zhang et al., 2018b) are often able to perform unidirectional but not bidirectional transforms between images $\boldsymbol{x}$ and texts $\boldsymbol{t}$. However, bidirectional transforms are straightforward for the proposed model, as $\boldsymbol{z}$ can be either drawn from the image encoder $q(\boldsymbol{z} \,|\, \boldsymbol{x})$ in (6), or drawn with an upward-downward Gibbs sampler (Zhou et al., 2016) from the conditional posteriors $p(\boldsymbol{z} \,|\, \boldsymbol{t})$ of the PGBN text decoder $p(\boldsymbol{t} \,|\, \boldsymbol{z})$ in (5).

## 2.3 VHE-RASTER-SCAN-GAN WITH A HIERARCHICAL-SEMANTIC MULTI-RESOLUTION IMAGE GENERATOR

While we find that VHE-StackGAN++ has already achieved impressive results, its simple concatenation of $\boldsymbol{\theta}^{(l)}$ does not fully exploit the semantically-meaningful hierarchical latent representation of the PGBN-based text decoder. For three DAG subnets inferred from three different datasets, as shown in Figs. 21 -23 of Appendix C.7, the higher-layer PGBN topics match general visual concepts, such as those on shapes, colors, and backgrounds, while the lower-layer ones provide finer details. This motivates us to develop an image generator to exploit the semantic structure, which matches coarse-to-fine visual concepts, to gradually refine its generation. To this end, as shown in Fig. 1(c), we develop "raster-scan" GAN that performs generation not only in a multi-scale low-to-high-resolution manner in each layer, but also a hierarchical-semantic coarse-to-fine fashion across layers.

Suppose we are building a three-layer raster-scan GAN to generate an image of size $256^2$. We randomly select an image $\boldsymbol{x}_n$ and then sample $\{\boldsymbol{\theta}_n^{(l)}\}_{l=1}^3$ from the variational posterior $\prod_{l=1}^3 q(\boldsymbol{\theta}_n^{(l)} \,|\, \boldsymbol{x}_n, \boldsymbol{\Phi}^{(l+1)}, \boldsymbol{\theta}_n^{(l+1)})$. First, the top-layer latent variable $\boldsymbol{\theta}^{(3)}$, often capturing general

semantic information, is transformed to hidden features $h_i^{(3)}$ for the $i^{th}$ branch:

$$h_1^{(3)} = F_1^{(3)}(\boldsymbol{\theta}^{(3)}); \;\; h_i^{(3)} = F_i^{(3)}(h_{i-1}^{(3)}, \boldsymbol{\theta}^{(3)}), \;\; i = 2, 3, \tag{7}$$

where $F_i^{(l)}$ is a CNN. Second, having obtained $\{h_i^{(3)}\}_{i=1}^3$, generators $\{G_i^{(3)}\}_{i=1}^3$ synthesize low-to-high-resolution image samples $\{\boldsymbol{s}_i^{(3)} = G_i^{(3)}(h_i^{(3)})\}_{i=1}^3$, where $\boldsymbol{s}_1^{(3)}$, $\boldsymbol{s}_2^{(3)}$, and $\boldsymbol{s}_3^{(3)}$ are of $16^2$, $32^2$, and $64^2$, respectively. Third, $\boldsymbol{s}_3^{(3)}$ is down-sampled to $\hat{\boldsymbol{s}}_3^{(3)}$ of size $32^2$ and combined with the information from $\boldsymbol{\theta}^{(2)}$ to provide the hidden features at layer two:

$$h_1^{(2)} = C(F_1^{(2)}(\boldsymbol{\theta}^{(2)}), \hat{\boldsymbol{s}}_3^{(3)}); \;\; h_i^{(2)} = F_i^{(2)}(h_{i-1}^{(2)}, \boldsymbol{\theta}^{(2)}), \;\; i = 2, 3, \tag{8}$$

where $C$ denotes concatenation along the channel. Fourth, the generators synthesize image samples $\{\boldsymbol{s}_i^{(2)} = G_i^{(2)}(h_i^{(2)})\}_{i=1}^3$, where $\boldsymbol{s}_1^{(2)}$, $\boldsymbol{s}_2^{(2)}$, and $\boldsymbol{s}_3^{(2)}$ are of $32^2$, $64^2$, and $128^2$, respectively. The same process is then replicated at layer one to generate $\{\boldsymbol{s}_i^{(1)} = G_i^{(1)}(h_i^{(1)})\}_{i=1}^3$, where $\boldsymbol{s}_1^{(1)}$, $\boldsymbol{s}_2^{(1)}$, and $\boldsymbol{s}_3^{(1)}$ are of size $64^2$, $128^2$, and $256^2$, respectively, and $\boldsymbol{s}_3^{(1)}$ becomes a desired high-resolution synthesized image with fine details. The detailed structure of raster-scan-GAN is described in Fig. 26 of Appendix D.3. PyTorch code is provided to aid the understanding and help reproduce the results.

Different from many existing methods (Gomez et al., 2017; Reed et al., 2016; Xu et al., 2018; Zhang et al., 2017b) whose textual feature extraction is separated from the end task, VHE-raster-scan-GAN performs joint optimization. As detailedly described in the Algorithm in Appendix E, at each mini-batch based iteration, after updating $\boldsymbol{\Phi}$ by the topic-layer-adaptive stochastic gradient Riemannian (TLASGR) MCMC of Cong et al. (2017), a Weibull distribution based reparameterization gradient (Zhang et al., 2018a) is used to end-to-end optimize the following objective:

$$\begin{aligned} \min_{\{G_i^{(l)}\}_{i,l}, \, \boldsymbol{\Omega}} \max_{\{D_i^{(l)}\}_{i,l}} \; &\mathbb{E}_{p_{\text{data}}(\boldsymbol{x}_n, \boldsymbol{t}_n)} \mathbb{E}_{\prod_{l=1}^3 q(\boldsymbol{\theta}_n^{(l)} \mid \boldsymbol{x}_n, \boldsymbol{\Phi}^{(l+1)}, \boldsymbol{\theta}_n^{(l+1)})} \big\{ -\log p(\boldsymbol{t}_n \mid \boldsymbol{\Phi}^{(1)}, \boldsymbol{\theta}_n^{(1)}) \\ &+ \textstyle\sum_{l=1}^3 \mathrm{KL}[q(\boldsymbol{\theta}_n^{(l)} \mid \boldsymbol{x}_n, \boldsymbol{\Phi}^{(l+1)}, \boldsymbol{\theta}_n^{(l+1)}) \,||\, p(\boldsymbol{\theta}_n^{(l)} \mid \boldsymbol{\Phi}^{(l+1)}, \boldsymbol{\theta}_n^{(l+1)})] \\ &+ \textstyle\sum_{l=1}^3 \sum_{i=1}^3 [\log D_i^{(l)}(\boldsymbol{x}_{n,i}^{(l)}, \boldsymbol{\theta}_n^{(l)}) + \log(1 - D_i^{(l)}(G_i^{(l)}(\boldsymbol{\theta}_n^{(l)}), \boldsymbol{\theta}_n^{(l)}))] \big\}, \end{aligned} \tag{9}$$

where $\{\boldsymbol{x}_{n,i}^{(l)}\}_{i=1,l=1}^{3,3}$ denote different resolutions of $\boldsymbol{x}_n$, corresponding to $\{\boldsymbol{s}_{n,i}^{(l)}\}_{i=1,l=1}^{3,3}$.

### 2.4 RELATED WORK ON JOINT IMAGE-TEXT LEARNING

Gomez et al. (2017) develop a CNN to learn a transformation from images to textual features pre-extracted by LDA. GANs have been exploited to generate images given pre-learned textual features extracted by RNNs (Denton et al., 2015; Reed et al., 2016; Zhang et al., 2017a; Xu et al., 2018; Zhang et al., 2018b; Li et al., 2019). All these works need a pre-trained linguistic model based on large-scale extra text data and the transformations between images and texts are only unidirectional. The recently proposed Obj-GAN (Li et al., 2019) needs even more side information such as the locations and labels of objects inside images, which could be difficult and costly to acquire in practice. On the other hand, probabilistic graphical model based methods (Srivastava & Salakhutdinov, 2012b;a; Wang et al., 2018) are proposed to learn a joint latent space for images and texts to realize bidirectional transformations, but their image generators are often limited to generating low-level image features. By contrast, VHE-raster-scan-GAN performs bidirectional end-to-end learning to capture and relate hierarchical visual and semantic concepts across multiple stochastic layers, capable of a wide variety of joint image-text learning and generation tasks, as described below.

## 3 EXPERIMENTAL RESULTS

For joint image-text learning, following previous work, we evaluate the proposed VHE-StackGAN++ and VHE-raster-scan-GAN on three datasets: CUB (Wah et al., 2011), Flower (Nilsback & Zisserman, 2008), and COCO (Lin et al., 2014), as described in Appendix F. Besides the usual text-to-image generation task, due to the distinct bidirectional inference capability of the proposed models, we can perform a rich set of additional tasks such as image-to-text, image-to-image, and noise-to-image-text-pair generations. Due to space constraint, we present below some representative results, and defer additional ones to the Appendix. We provide the details of our experimental settings in Appendix F. PyTorch code is provided at https://github.com/BoChenGroup/VHE-GAN.

Table 1: Inception score (IS, larger is better) and Frechet inception distance (FID, smaller is better) of StackGAN++ (Zhang et al., 2017b), HDGAN (Zhang et al., 2018b), AttGAN (Xu et al., 2018), Obj-GAN (Li et al., 2019), and the proposed VHE-raster-scan-GAN; the values labeled with $^*$ are calculated by the provided well-trained models and the others are quoted from the original publications; see Tab. 5 in Appendix C.1 for the error bars of IS. Note that while the FID of Obj-GAN is the lowest, it does not necessarily imply it produces high-quality images, as shown in Figs. 13 and 27; this is because FID only measures the similarity on the image feature space, but ignores the shapes of objects and diversity of generated images. More discussions can be found in Section 3.1 and Appendix G.

| Method | StackGAN++ | | HDGAN | | AttnGAN | | Obj-GAN | | VHE-raster-scan-GAN | |
|--------|-----|-----|-----|-----|-----|-----|-----|-----|-----|-----|
| Criterion | IS | FID | IS | FID | IS | FID | IS | FID | IS | FID |
| Flower | 3.26 | 48.68 | 3.45 | 40.12$^*$ | – | – | - | - | **3.72** | **35.13** |
| CUB | 3.84 | 15.30 | 4.15 | 13.48$^*$ | 4.36 | 13.02$^*$ | - | - | **4.41** | **12.02** |
| COCO | 8.30 | 81.59 | 11.86 | 78.16$^*$ | 25.89 | 77.01$^*$ | 26.58$^*$ | **36.98**$^*$ | 27.16 | 75.88 |

Table 2: Ablation study for image-to-text learning, where the structures of different variations of raster-scan-GAN are illustrated in Figs. 1(b), 1(d), and 1(e).

| Method | PGBN+StackGAN++ | | VHE-vanilla-GAN | | VHE-StackGAN++ | | VHE-simple-raster-scan-GAN | |
|--------|-----|-----|-----|-----|-----|-----|-----|-----|
| Criterion | IS | FID | IS | FID | IS | FID | IS | FID |
| Flower | 3.29 | 41.04 | 3.01 | 52.15 | 3.56 | 38.66 | 3.62 | 36.18 |
| CUB | 3.92 | 13.79 | 3.52 | 21.24 | 4.20 | 12.93 | 4.31 | 12.35 |
| COCO | 10.63 | 79.65 | 6.36 | 97.15 | 12.63 | 78.02 | 20.13 | 77.18 |

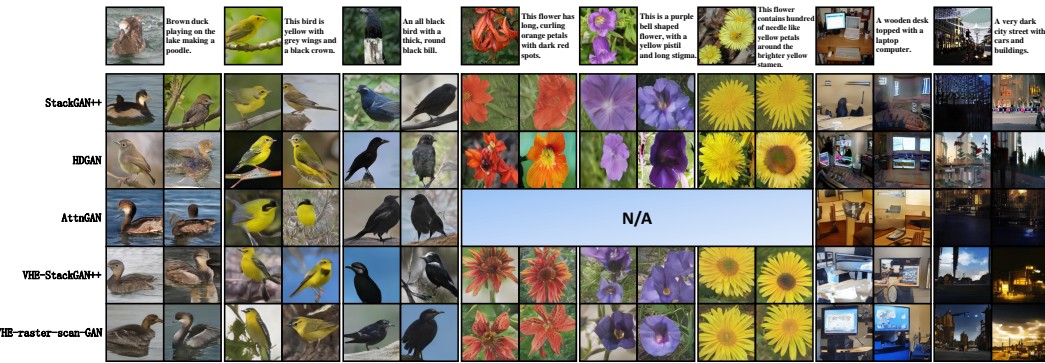

Figure 2: Comparison on image generation given texts from CUB, Flower, and COCO. Shown in the top row are the textual descriptions and their associated real images; see Appendix C.2 for higher-resolution images. Note AttnGAN did not perform experiments on Flower and hence its results on Flower are not shown, and since Obj-GAN only performed experiment on COCO, we defer its visual results to Appendix C.3.

## 3.1 TEXT-TO-IMAGE LEARNING

Although the proposed VHE-GANs do not have a text encoder to directly project a document to the shared latent space, given a document and a set of topics inferred during training, we use the upward-downward Gibbs sampler of Zhou et al. (2016) to draw $\{\boldsymbol{\theta}^{(l)}\}_{l=1}^{L}$ from its conditional posterior under PGBN, which are then fed into the GAN image generator to synthesize random images.

**Text-to-image generation:** In Tab. 1, with inception score (IS) (Salimans et al., 2016) and Frechet inception distance (FID) (Heusel et al., 2017), we compare our models with three state-of-the-art GANs in text-to-image generation. For visualization, we show in the top row of Fig. 2 different test textual descriptions and the real images associated with them, and in the other rows random images generated conditioning on these textual descriptions by different algorithms. Higher-resolution images are shown in Appendix C.2. We also provide example results on COCO, a much more challenging dataset, in Fig. 13 of Appendix C.3.

It is clear from Fig. 2 that although both StackGAN++ (Zhang et al., 2017b) and HDGAN (Zhang et al., 2018b) generate photo-realistic images nicely matched to the given texts, they often misrepresent or ignore some key textual information, such as "black crown" for the 2nd test text, "yellow pistil" for 5th, "yellow stamen" for 6th, and "computer" for 7th. These observations also apply to AttnGAN (Xu et al., 2018). By contrast, both the proposed VHE-StackGAN++ and VHE-raster-scan-GAN do a better job in capturing and faithfully representing these key textual information into their generated images. Fig. 13 for COCO further shows the advantages of VHE-raster-scan-GAN in better

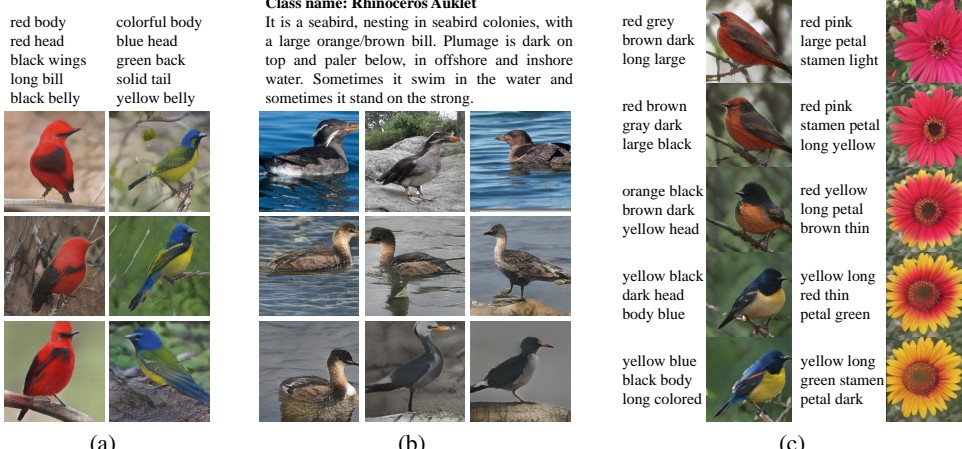

Figure 3: Example results of VHE-raster-scan-GAN on three different tasks: (a) image generation given five textual attributes; (b) image generation given a long class-specific document (showing three representative sentences for brevity) from CUB; and (c) latent space interpolation for joint image-text generation on CUB (left column) and Flower (right column), where the texts in the first and last row are given.

representing the given textual information in its generated images. Note Obj-GAN, which learns a bounding box generator that restricts object locations, obtains the lowest FID on COCO. However, it appears that this type of restriction significantly improves FID at the expense of sacrificing the diversity of generated images given text, as shown in Fig. 27 of Appendix G. From the results in Fig. 13, it also appears that Obj-GAN overly emphasizes correctly arranging the spatial locations of different visual features, which is important to achieve low FID, but does not do well in generating correct object shapes, which is important to visual effect. Besides, the training of Obj-GAN requires more side information including the locations and labels of objects in the images, which are often not provided in practice (e.g., neither CUB nor Flower comes with this type of side information). While the proposed VHE-GAN models do not need these additional side information, they could be further improved by following Obj-GAN to take them into consideration.

As discussed in Section 2.2, compared with sequence models, topic models can be applied to more diverse textual descriptions, including textual attributes and long documents. For illustration, we show in Figs. 3(a) and 3(b) example images generated conditioning on a set of textual attributes and an encyclopedia document, respectively. These synthesized images are photo-realistic and their visual contents well match the semantics of the given texts. Trained on CelebA (Liu et al., 2015), we provide in Fig. 9 examples of facial image generation given attributes; see Appendix B for details.

**Ablation studies:** We also consider several ablation studies for text-to-image generation, as shown in Tab. 2. **First**, we modify StackGAN++ (Zhang et al., 2017b), using the text features extracted by PGBN to replace the original ones by RNN, referred to as PGBN+StackGAN++. It is clear that PGBN+StackGAN++ outperforms the original StackGAN++, but underperforms VHE-StackGAN++, which can be explained by that 1) the PGBN deep topic model is more effective in extracting macro-level textual information, such as key words, than RNNs; and 2) jointly end-to-end training the textual feature extractor and image encoder, discriminator, and generator helps better capture and relate the visual and semantical concepts. **Second**, note that VHE-StackGAN++ has the same structured image generator as both StackGAN++ and HDGAN do, but performs better than them. We attribute its performance gain to 1) its PGBN deep topic model helps better capture key semantic information from the textual descriptions; and 2) it performs end-to-end joint image-text learning via the VHE-GAN framework, rather than separating the extraction of textual features from text-to-image generation. **Third**, VHE-vanilla-GAN underperforms VHE-StackGAN++, suggesting that the stacking structure is helpful for generating high resolution images, as previously verified in Zhang et al. (2017a). VHE-simple-raster-scan-GAN outperforms VHE-StackGAN++ but underperforms VHE-raster-scan-GAN, confirming the benefits of combining the stacking and raster-scan structures. More visual results for ablation studies can be found in Appendix C.2. Below we focus on illustrating the outstanding performance of VHE-raster-scan-GAN.

**Latent space interpolation:** In order to understand the jointly learned image and text manifolds, given texts $t_1$ and $t_2$, we draw $\theta_1$ and $\theta_2$ and use the interpolated variables between them to generate

both images via the GAN's image generator and texts via the PGBN text decoder. As in Fig. 3(c), the first row shows the true texts $t_1$ and images generated with $\theta_1$, the last row shows $t_2$ and images generated with $\theta_2$, and the second to fourth rows show the generated texts and images with the interpolations from $\theta_1$ to $\theta_2$. The strong correspondences between the generated images and texts, with smooth changes in colors, object positions, and backgrounds between adjacent rows, suggest that the latent space of VHE-raster-scan-GAN is both visually and semantically meaningful. Additional more fine-gridded latent space interpolation results are shown in Figs. 15-18 of Appendix C.4.

**Visualization of captured semantic and visual concepts:** Zhou et al. (2016) show that the semantic concepts extracted by PGBN and their hierarchical relationships can be represented as a DAG, only a subnet of which will be activated given a specific text input. In each subplot of Fig. 4, we visualize example topic nodes of the DAG subnet activated by the given text input, and show the corresponding images generated at different hidden layers. There is a good match at each layer between the visual contents of the generated images and semantics of the top activated topics, which are mainly about general shapes, colors, or backgrounds at the top layer, and become more and more fine-grained when moving downward. In Fig. 5, for the DAG learned on COCO, we show a representative subnet that is rooted at a top-layer node about "rooms and objects at home," and provide both semantic and visual representations for each node. Being able to capture and relate hierarchical semantic and visual concepts helps explain the state-of-the-art performance of VHE-raster-scan-GAN.

## 3.2 IMAGE-TO-TEXT LEARNING

VHE-raster-scan-GAN can perform a wide variety of extra tasks, such as image-to-text generation, text-based zero-shot learning (ZSL), and image retrieval given a text query. In particular, given image $x_n$, we draw $\hat{t}_n$ as $\hat{t}_n \mid \theta_n \sim p(t \mid \Phi, \theta_n), \ \theta_n \mid x_n \sim q_\Omega(\theta \mid \Phi, x_n)$ and use it for downstream tasks.

**Image-to-text generation:** Given an image, we may generate some key words, as shown in Fig. 6(a), where the true and generated ones are displayed on the left and right of the input image, respectively. It is clear that VHE-raster-scan-GAN successfully captures the object colors, shapes, locations, and backgrounds to predict relevant key words.

**Text-based ZSL:** Text-based ZSL is a specific task that learns a relationship between images and texts on the seen classes and transfer it to the unseen ones (Fu et al., 2018). We follow the the same settings on CUB and Flower as existing text-based ZSL methods summarized in Tab. 3. There are two default splits for CUB—the hard (CUB-H) and easy one (CUB-E)—and one split setting for Flower, as described in Appendix F. Note that except for our models that infer a shared semantically meaningful latent space between two modalities, none of the other methods have generative models for both modalities, regardless of whether they learn a classifier or a distance metric in a latent space for ZSL. Tab. 3 shows that VHE-raster-scan-GAN clearly outperforms the state of the art in terms of the Top-1 accuracy on both the CUB-H and Flower, and is comparable to the second best on CUB-E (it is the best among all methods that have reported their Top-5 accuracies on CUB-E). Note for CUB-E, every unseen class has some corresponding seen classes under the same super-category, which makes the classification of surface or distance metric learned on the seen classes easier to generalize to the unseen ones. We also note that both GAZSL and ZSLPP rely on visual part detection to extract image features, making their performance sensitive to the quality of the visual part detector that often has to be elaborately tuned for different classes and hence limiting their generalization ability, for example, the visual part detector for birds is not suitable for flowers. Tab. 3 also includes the results of ZSL using VHE, which show that given the same structured text decoder and image encoder, VHE consistently underperforms both VHE-StackGAN++ and VHE-raster-scan-GAN. This suggests 1) the advantage of a joint generation of two modalities, and 2) the ability of GAN in helping VHE achieve better data representation. The results in Tab. 3 also show that the ZSL performance of VHE-raster-scan-GAN has a clear trend of improvement as PGBN becomes deeper, suggesting the advantage of having a multi-stochastic-hidden-layer deep topic model for text generation. We also collect the ZSL results of the last 1000 mini-batch based stochastic gradient update iterations to calculate the error bars. For existing methods, since there are no error bars provided in published paper, we only provide the text error bars of the methods that have publicly accessible code.

## 3.3 IMAGE/TEXT RETRIEVAL

As discussed in Section 2.4, the proposed models are able to infer the shared latent space given either an image or text. We test both VHE-StackGAN++ and VHE-raster-scan-GAN on the same image/text

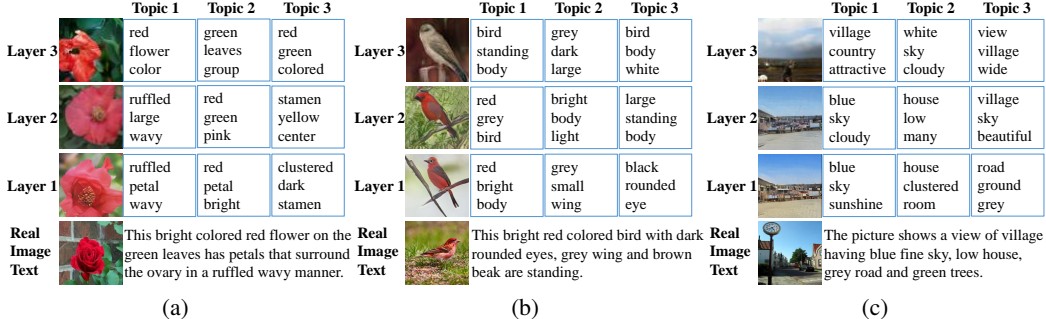

Figure 4: Visualization of example semantic and visual concepts captured by a three-stochastic-hidden-layer VHE-raster-scan-GAN from (a) Flower, (b) Bird, and (c) COCO. In each subplot, given the real text $t_n$ shown at the bottom, we draw $\{\theta_n^{(l)}\}_{l=1}^3$ via Gibbs sampling; we show the three most active topics in $\Phi^{(l)}$ (ranked by the weights of $\theta_n^{(l)}$) at layer $l = 3, 2, 1$, where each topic is visualized by its top three words; and we feed $\{\theta_n^{(l)}\}_{l=1}^3$ into raster-scan-GAN to generate three random images (one per layer, coarse to fine from layers 3 to 1).

Table 3: Accuracy (%) of ZSL on CUB and Flower. Note that some of them are attribute-based methods but applicable in our setting by replacing attribute vectors with text features (labeled by $^*$), as discussed in (Elhoseiny et al., 2017b).

| Text-ZSL dataset | CUB-H | CUB-E | | Flower |
|---|---|---|---|---|
| Accuracy criterion | top-1 | top-1 | top-5 | top-1 |
| WAC-Kernel (Elhoseiny et al., 2017a) | $7.7 \pm 0.28$ | $33.5 \pm 0.22$ | $64.3 \pm 0.20$ | $9.1 \pm 2.77$ |
| ZSLNS (Qiao et al., 2016) | $7.3 \pm 0.36$ | $29.1 \pm 0.28$ | $61.8 \pm 0.22$ | $8.7 \pm 2.46$ |
| ESZSL$^*$ (Romeraparedes & Torr, 2015) | $7.4 \pm 0.31$ | $28.5 \pm 0.26$ | $59.9 \pm 0.20$ | $8.6 \pm 2.53$ |
| SynC$^*$ (Changpinyo et al., 2016) | 8.6 | 28.0 | 61.3 | 8.2 |
| ZSLPP (Elhoseiny et al., 2017b) | 9.7 | 37.2 | – | – |
| GAZSL (Zhu et al., 2018) | $10.3 \pm 0.26$ | $\mathbf{43.7 \pm 0.28}$ | $67.61 \pm 0.24$ | – |
| VHE-L3 | $14.0 \pm 0.24$ | $34.6 \pm 0.25$ | $64.6 \pm 0.20$ | $8.9 \pm 1.57$ |
| VHE-StackGAN++-L3 | 16.1 | 38.5 | 68.2 | 10.6 |
| VHE-raster-scan-GAN-L1 | $11.7 \pm 0.31$ | $32.1 \pm 0.32$ | $62.6 \pm 0.33$ | $9.4 \pm 1.68$ |
| VHE-raster-scan-GAN-L2 | $14.9 \pm 0.26$ | $37.1 \pm 0.24$ | $64.6 \pm 0.25$ | $11.0 \pm 1.54$ |
| VHE-raster-scan-GAN-L3 | $\mathbf{16.7 \pm 0.24}$ | $39.6 \pm 0.20$ | $\mathbf{70.3 \pm 0.18}$ | $\mathbf{12.1 \pm 1.47}$ |

Table 4: Comparison of the image-to-text retrieval performance, measured by Top-1 accuracy, and text-to-image retrieval performance, measured by AP@50, between different methods on CUB-E.

| Method | CNN-LSTM (Li et al., 2017) | AttnGAN (Xu et al., 2018) | TA-GAN (Nam et al., 2018) | VHE-StackGAN++ | VHE-raster-scan-GAN |
|---|---|---|---|---|---|
| Top1-ACC(%) | 61.5 | 55.1 | 61.3 | 60.2 | **61.7** |
| AP@50(%) | 57.6 | 51.0 | **62.8** | 61.3 | 62.6 |

retrieval tasks as in TA-GAN (Nam et al., 2018), where we use the cosine distance between the inferred latent space given images ($q(\theta \mid x)$, image encoder) and these given texts ($p(\theta \mid t)$, Gibbs sampling) to compute the similarity scores. Similar with TA-GAN, the top-1 image-to-text retrieval accuracy (Top-1 Acc) and the percentage of matching images in top-50 text-to-image retrieval results (AP@50) on CUB-E dataset are used to measure the performance. As shown in Table 4, VHE-raster-scan-GAN clearly outperforms AttnGAN (Xu et al., 2018) and is comparable with TA-GAN. Note TA-GAN needs to extract its text features based on the fastText model (Bojanowski et al., 2017) pre-trained on a large corpus, while VHE-raster-scan-GAN learns everything directly from the current dataset in an end-to-end manner. Also, VHE-raster-scan-GAN outperforms VHE-StackGAN++, which further confirms the benefits of combining both the stacking and raster scan structures.

## 3.4 GENERATION OF RANDOM TEXT-IMAGE PAIRS

Below we show how to generate data samples that contain both modalities. After training a three-stochastic-hidden-layer VHE-raster-scan-GAN, following the data generation process of the PGBN text decoder, given $\{\Phi^{(l)}\}_{l=1}^3$ and $r$, we first generate $\theta^{(3)} \sim \text{Gam}\left(r, 1/s^{(4)}\right)$ and then downward propagate it through the PGBN as in (5) to calculate the Poisson rates for all words using $\Phi^{(1)}\theta^{(1)}$. Given a random draw, $\{\theta^{(l)}\}_{l=1}^3$ is fed into the raster-scan-GAN image generator to generate a

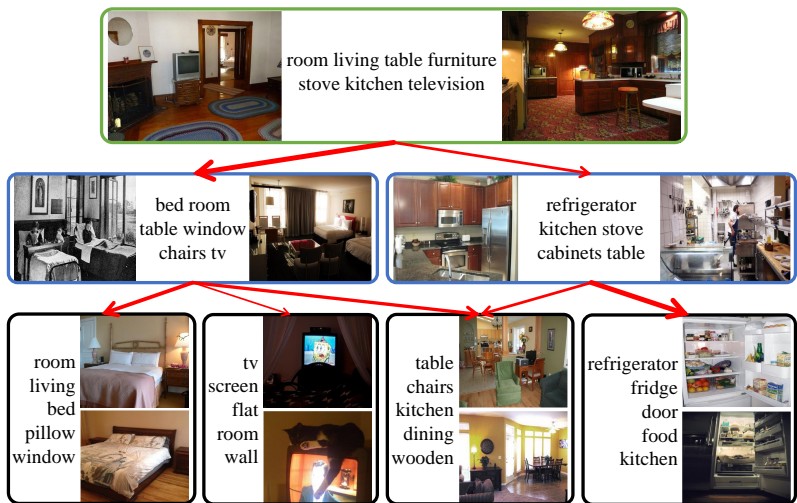

Figure 5: An example topic hierarchy learned on COCO and its visual representation. We sample $\boldsymbol{\theta}_n^{(1:3)} \sim q(\boldsymbol{\theta}_n^{(1:3)} \mid \boldsymbol{\Phi}, \boldsymbol{x}_n)$ for all $n$; for topic node $k$ of layer $l$, we show both its top words and the top two images ranked by their activations $\theta_{nk}^{(l)}$.

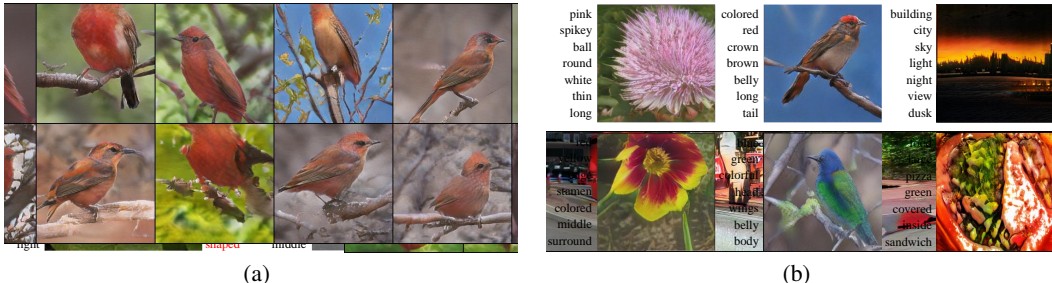

Figure 6: Example results of using VHE-raster-scan-GAN for (a) image-to-textual-tags generation, where the generated tags highlighted in red are included in the original ones; (b) image-text-pair generations (columns from left to right are based on Flower, CUB, and COCO, respectively).

corresponding image. Shown in Fig. 6(b) are six random draws, for each of which we show its top seven words and generated image, whose relationships are clearly interpretable, suggesting that VHE-raster-scan-GAN is able to recode the key information of both modalities and the relationships between them. In addition to the tasks shown above, VHE-raster-scan-GAN can also be used to perform image retrieval given a text query, and image regeneration; see Appendices C.5 and C.6 for example results on these additional tasks.

## 4 CONCLUSION

We develop variational hetero-encoder randomized generative adversarial network (VHE-GAN) to provide a plug-and-play joint image-text modeling framework. VHE-GAN is a versatile deep generative model that integrates off-the-shelf image encoders, text decoders, and GAN image discriminators and generators into a coherent end-to-end learning objective. It couples its VHE and GAN components by feeding the VHE variational posterior in lieu of noise as the source of randomness of the GAN generator. We show VHE-StackGAN++ that combines the Poisson gamma belief network, a deep topic model, and StackGAN++ achieves competitive performance, and VHE-raster-scan-GAN, which further improves VHE-StackGAN++ by exploiting the semantically-meaningful hierarchical structure of the deep topic model, generates photo-realistic images not only in a multi-scale low-to-high-resolution manner, but also in a hierarchical-semantic coarse-to-fine fashion, achieving outstanding results in many challenging image-to-text, text-to-image, and joint text-image learning and generation tasks.

ACKNOWLEDGEMENTS

B. Chen acknowledges the support of the Program for Young Thousand Talent by Chinese Central Government, the 111 Project (No. B18039), NSFC (61771361), NSFC for Distinguished Young Scholars (61525105), Shaanxi Innovation Team Project, and the Innovation Fund of Xidian University. M. Zhou acknowledges the support of the U.S. National Science Foundation under Grant IIS-1812699.

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

# A   MODEL PROPERTY OF VHE-GAN AND RELATED WORK

Let us denote $q(\boldsymbol{z}) = \mathbb{E}_{\boldsymbol{x} \sim p_{\text{data}}(\boldsymbol{x})}[q(\boldsymbol{z} \,|\, \boldsymbol{x})] = \frac{1}{N} \sum_{n=1}^{N} q(\boldsymbol{z} \,|\, \boldsymbol{x}_n)$ as the aggregated posterior (Hoffman & Johnson, 2016; Makhzani et al., 2015). Removing the triple-data-reuse training strategy, we can re-express the VHE-GAN objective in (4) as

$$\min_{E, G_{\text{vae}}, G_{\text{gan}}} \max_{D} [-\text{ELBO}_{\text{vhe}} + \mathcal{L}_{\text{gan}}], \mathcal{L}_{\text{gan}} := \mathbb{E}_{\boldsymbol{x} \sim p_{\text{data}}(\boldsymbol{x})} \ln D(\boldsymbol{x}) + \mathbb{E}_{\boldsymbol{z} \sim q(\boldsymbol{z})} \ln(1 - D(G_{\text{gan}}(\boldsymbol{z}))),$$
(10)

which corresponds to a naive combination of the VHE and GAN training objectives, where the data samples used to train the VHE, GAN generator, and GAN discriminator in each gradient update iteration are not imposed to be the same. While the naive objective function in (10) differs from the true one in (4) that is used to train VHE-GAN, it simplifies the analysis of its theoretical property, as described below.

Let us denote $q(\boldsymbol{z}, \boldsymbol{x}, \boldsymbol{t}) := q(\boldsymbol{z} \,|\, \boldsymbol{x}) p_{data}(\boldsymbol{x}, \boldsymbol{t})$ as the joint distribution of $(\boldsymbol{x}, \boldsymbol{t})$ and $\boldsymbol{z}$ under the VHE variational posterior $q(\boldsymbol{z} \,|\, \boldsymbol{x})$, $I_q(\boldsymbol{x}, \boldsymbol{z}) := \mathbb{E}_{q(\boldsymbol{z}, \boldsymbol{x})} \big[ \ln \frac{q(\boldsymbol{z}, \boldsymbol{x})}{q(\boldsymbol{z}) p_{data}(\boldsymbol{x})} \big]$ as the mutual information between $\boldsymbol{x} \sim p_{data}(\boldsymbol{x})$ and $\boldsymbol{z} \sim q(\boldsymbol{z})$, and $\text{JDS}(p_1 || p_2) := \frac{1}{2} \text{KL}[p_1 || (p_1 + p_2)/2] + \frac{1}{2} \text{KL}[p_2 || (p_1 + p_2)/2]$ as the Jensen–Shannon divergence between distributions $p_1$ and $p_2$. Similar to the analysis in Hoffman & Johnson (2016), the VHE's ELBO can be rewritten as $\text{ELBO}_{\text{vhe}} = \mathbb{E}_{q(\boldsymbol{z}, \boldsymbol{x}, \boldsymbol{t})} [\log p(\boldsymbol{t} \,|\, \boldsymbol{z})] - I_q(\boldsymbol{x}, \boldsymbol{z}) - \text{KL}[q(\boldsymbol{z}) || p(\boldsymbol{z})]$, where the mutual information term can also be expressed as $I_q(\boldsymbol{x}, \boldsymbol{z}) = \mathbb{E}_{\boldsymbol{x} \sim p_{data}(\boldsymbol{x})} \text{KL}[q(\boldsymbol{z} \,|\, \boldsymbol{x}) || q(\boldsymbol{z})]$. Thus maximizing the ELBO encourages the mutual information term $I_q(\boldsymbol{x}, \boldsymbol{z})$ to be minimized, which means while the data reconstruction term $\mathbb{E}_{q(\boldsymbol{z}, \boldsymbol{x}, \boldsymbol{t})} [\log p(\boldsymbol{t} \,|\, \boldsymbol{z})]$ needs to be maximized, part of the VHE optimization objective penalizes a $\boldsymbol{z}$ from carrying the information of the $\boldsymbol{x}$ that it is encoded from. This mechanism helps provide necessary regularization to prevent overfitting. As in Goodfellow et al. (2014), with an optimal discriminator $D_G^*$ for generator $G$, we have $\min \mathcal{L}_{\text{GAN}}(D_G^*, G) = \ln 4 + 2\text{JSD}(p_{data}(\boldsymbol{x}) || p_{G_{\boldsymbol{z}}}(\boldsymbol{x}))$, where $p_{G_{\boldsymbol{z}}(\boldsymbol{x})}$ denotes the distribution of the generated data $G(\boldsymbol{z})$ that use $\boldsymbol{z} \sim q(\boldsymbol{z})$ as the random source fed into the GAN generator. The JSD term is minimized when $p_{G_{\boldsymbol{z}}}(\boldsymbol{x}) = p_{data}(\boldsymbol{x})$.

With these analyses, given an optimal GAN discriminator, the naive VHE-GAN objective function in (10) reduces to

$$\min_{E, G_{\text{gan}}, G_{\text{vae}}} -\mathbb{E}_{q(\boldsymbol{z}, \boldsymbol{x}, \boldsymbol{t})} [\log p(\boldsymbol{t} \,|\, \boldsymbol{z})] + \text{KL}[q(\boldsymbol{z}) || p(\boldsymbol{z})] + I_q(\boldsymbol{x}, \boldsymbol{z}) + 2\text{JSD}(p_{data}(\boldsymbol{x}) || p_{G_{\boldsymbol{z}}}(\boldsymbol{x})). \quad (11)$$

From the VHEs' point of view, examining (11) shows that it alleviates the inherent conflict in VHE of maximizing the ELBO and maximizing the mutual information $I_q(\boldsymbol{x}, \boldsymbol{z})$. This is because while the VHE part of VHE-GAN still relies on minimizing $I_q(\boldsymbol{x}, \boldsymbol{z})$ to regularize the learning, the GAN part tries to transform $q(\boldsymbol{z})$ through the GAN generator to match the true data distribution $p_{data}(\boldsymbol{x})$. In other words, while its VHE part penalizes a $\boldsymbol{z}$ from carrying the information about the $\boldsymbol{x}$ that it is encoded from, its GAN part encourages a $\boldsymbol{z}$ to carry information about the true data distribution $p_{data}(\boldsymbol{x})$, but not necessarily the observed $\boldsymbol{x}$ that it is encoded from.

From the GANs' point of view, examining (11) shows that it provides GAN with a meaningful latent space, necessary for performing inference and data reconstruction (with the aid of the data-triple-use training strategy). More specifically, this latent representation is also used by the VHE to maximize the data log-likelihood, a training procedure that tries to cover all modes of the empirical data distribution rather than dropping modes. For VHE-GAN (4), the source distribution is $q(\boldsymbol{z} \,|\, \boldsymbol{x})$, not only allowing GANs to participate in posterior inference and data reconstruction, but also helping GANs resist mode collapse. In the following, we discuss some related works on combining VAEs and GANs.

## A.1   RELATED WORK ON COMBINING VAEs AND GANs

Examples in improving VAEs with adversarial learning include Mescheder et al. (2017), which allows the VAEs to take implicit encoder distribution, and adversarial auto-encoder (Makhzani et al., 2015) and Wasserstein auto-encoder (Tolstikhin et al., 2018), which drop the mutual information term from the ELBO and use adversarial learning to match the aggregated posterior and prior. Examples in allowing GANs to perform inference include Dumoulin et al. (2017) and Donahue et al. (2017),

which use GANs to match the joint distribution $q(\boldsymbol{z} \,|\, \boldsymbol{x})p_{data}(\boldsymbol{x})$ defined by the encoder and the one $p(\boldsymbol{x} \,|\, \boldsymbol{z})p(\boldsymbol{z})$ defined by the generator. However, they often do not provide good data reconstruction. Examples in using VAEs or maximum likelihood to help GANs resist mode collapse include Che et al. (2017); Srivastava et al. (2017); Grover et al. (2018). Another example is VAEGAN (Larsen et al., 2016) that combines unit-wise likelihood at hidden layer and adversarial loss at original space, but its update of the encoder is separated from the GAN mini-max objective. On the contrary, IntroVAE (Huang et al., 2018) retains the pixel-wise likelihood with an adversarial regularization on the latent space. Sharing network between the VAE decoder and GAN generator in VAEGAN and IntroVAE, however, limit them to model a single modality.

## B  MORE DISCUSSION ON SEQUENCE MODELS AND TOPIC MODELS IN TEXT ANALYSIS.

In Section 3.1, we have discussed two models to represent the text: sequence models and topic models. Considering the versatility of topic models (Wang et al., 2009; Jin et al., 2015; Zhou et al., 2016; Srivastava & Salakhutdinov, 2012a; 2014; Wang et al., 2018; Elhoseiny et al., 2017b; Zhu et al., 2018) in dealing with different types of textual information, and its effectiveness in capturing latent topics that are often directly related to macro-level visual information (Gomez et al., 2017; Dieng et al., 2017; Lau et al., 2017), we choose a state-of-the-art deep topic model, PGBN, to model the textual descriptions in VHE. Due to space constraint, we only provide simple illustrations in Figs. 3(a) and 3(b). In this section, more insights and discussions are provided.

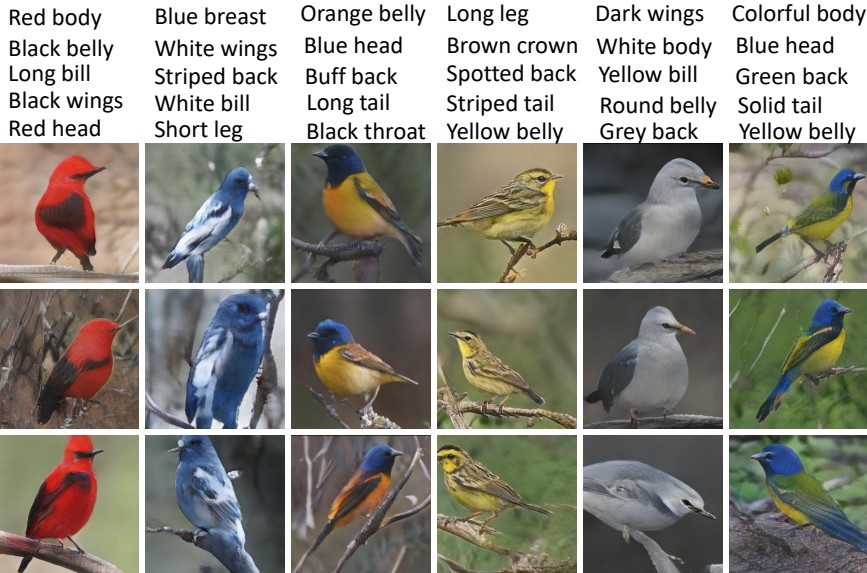

Figure 7: Generated random images by VHE-raster-scan-GAN conditioning on five binary attributes.

As discussed before, topic models are able to model non-sequential texts such as binary attributes. The CUB dataset provides 312 binary attributes (Wah et al., 2011) for each images, such as whether "crown color is blue" and whether "tail shape is solid" to define the color or shape of different body parts of a bird. We first transform these binary attributes for the $n$th image to a 312-dimensional binary vector $\boldsymbol{t}_n$, whose $i$th element is 1 or 0 depending on whether the bird in this image owns the $i$th attribute or not. The binary attribute vectors $\boldsymbol{t}_n$ are used together with the corresponding bird images $\boldsymbol{x}_n$ to train VHE-raster-scan-GAN. As shown in Fig. 7, we generate images given five binary attributes, which are formed into a 312-dimensional binary vector $\boldsymbol{t}$ (with five non-zero elements at these five attributes) that becomes the input to the PGBN text decoder. Clearly, these generated images are photo-realistic and faithfully represent the five provided attributes.

The proposed VHE-GANs can also well model long documents. In text-based ZSL discussed in Section 3.2, each class (not each image) is represented as a long encyclopedia document, whose global

semantic structure is hard to captured by existing sequence models. Besides a good ZSL performance achieved by VHE-raster-scan-GAN, illustrating its advantages of text generation given images, we show Fig. 8 example results of image generation conditioning on long encyclopedia documents on the unseen classes of CUB-E (Qiao et al., 2016; Akata et al., 2015) and Flower (Elhoseiny et al., 2017a).

**Class name: Rhinoceros Auklet**

It is a seabird, nesting in seabird colonies, with a large orange/brown bill. Plumage is dark on top and paler below, in offshore and inshore water. Sometimes it swim in the water and sometimes it stand on the strong.

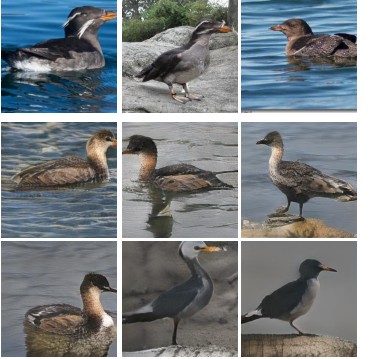

**Class name: Yellow Bellied Flycatcher**

Brownish-olive upperparts, darker on the wings and tail, yellowish underparts. Have small bill short tail, on a perch low or in the middle of a tree. Its eyes are dark and round with radiating vigor, like looking for food or insects.

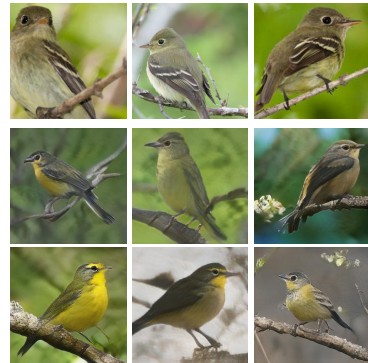

(a)

**Class name: Ball Moss**

It tends to form a spheroid shape ranging in size from a golf ball to a soccer ball. It may hinder tree growth. Its petals are stripe-like yellow ones and its stamen is also round dark brown or yellow.

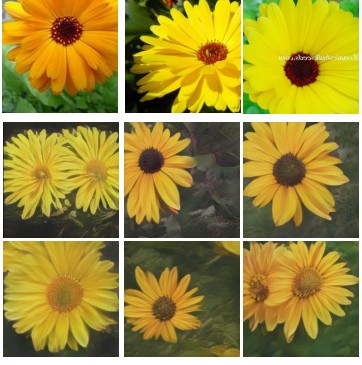

**Class name: Barberton Daisy**

It bear a large capitulum with striking, two-lipped ray floret in yellow or orange. Colors include white, yellow, and pink. Its petals are medium, and each of them is round and the number is about six.

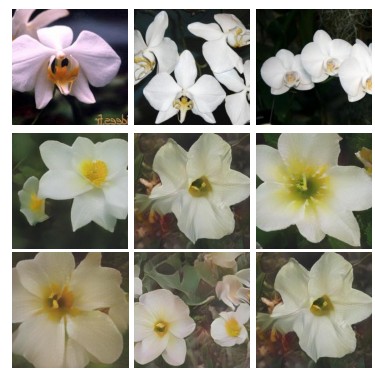

(b)

Figure 8: Image generation conditioning on long encyclopedia documents using VHE-raster-scan-GAN trained on (a) CUB-E and (b) Flower. Shown in the top part of each subplot are representative sentences taken from the long document that describes an unseen class; for the three rows of images shown in the bottom part, the first row includes three real images from the corresponding unseen class, and the other two rows include a total of six randomly generated images conditioning on the long encyclopedia document of the corresponding unseen class.

Analogous to how the Bird images are generated in Fig. 7, we also perform facial image generation given a set of textual attributes. On CelebA dataset, given attributes, we train VHE-stackGAN++ and

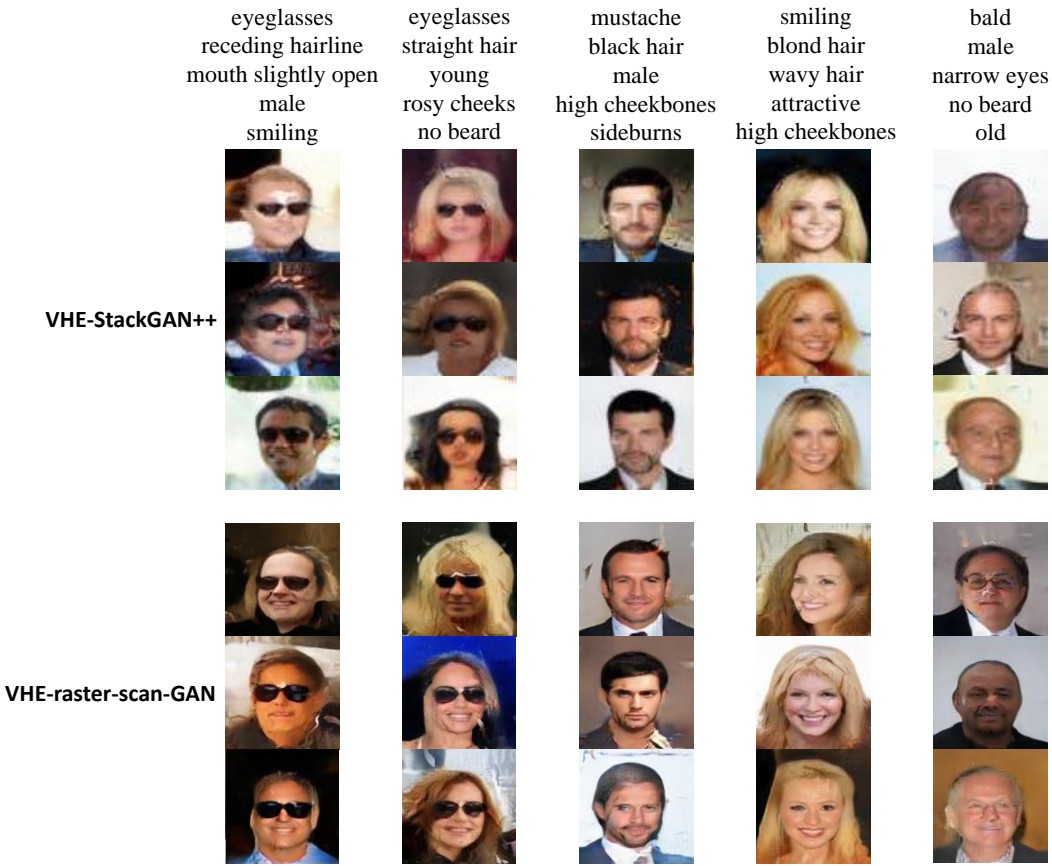

Figure 9: Example results of facial image generation conditioning on five textual attributes, by VHEStackGAN++ and VHE-raster-scan-GAN trained on the CelebA dataset. Both models are trained with 20 epochs, with the output resolution set as $128 \times 128$. Note our current network architecture, designed mainly for natural images, has not yet been fine-tuned for facial images.

VHE-raster-scan-GAN to generate the facial images with resolution $128 \times 128$. As shown in Fig. 9, after the training of 20 epochs, we generate facial images given five attributes. While the facial images generated by both models nicely match the given attributes, VHE-raster-scan-GAN provides higher visual quality and does a better job in representing the details.

# C MORE EXPERIMENTAL RESULTS ON JOINT IMAGE-TEXT LEARNING

## C.1 TABLES 1 AND 2 WITH ERROR BARS.

For text-to-image generation tasks, we use the official pre-defined training/testing split (illustrated in Appendix F) to train and test all the models. Following the definition of error bar of IS in StackGAN++ (Zhang et al., 2017b), HDGAN (Zhang et al., 2018b), and AttnGAN (Xu et al., 2018), we provide the IS results with error bars for various methods in Table 5, where the results of the StackGAN++ , HDGAN, and AttnGAN are quoted from the published papers. The FID error bar is not included as it has not been clearly defined.

Table 5: Inception score (IS) results in Table 1 with error bars.

| Method | StackGAN++ | HDGAN | AttnGAN | Obj-GAN | VHE-raster-scan-GAN |
|--------|------------|-------|---------|---------|---------------------|
| Flower | $3.26 \pm .01$ | $3.45 \pm .07$ | – | - | **$3.72 \pm .01$** |
| CUB | $3.84 \pm .06$ | $4.15 \pm .05$ | $4.36 \pm .03$ | - | **$4.41 \pm .03$** |
| COCO | $8.30 \pm .10$ | $11.86 \pm .18$ | $25.89 \pm .47$ | $26.68 \pm .52$ | **$27.16 \pm .23$** |

Table 6: Inception score (IS) results in Table 2 with error bars.

| Method | PGBN+StackGAN++ | VHE-vanilla-GAN | VHE-StackGAN++ | VHE-simple-raster-scan-GAN |
|--------|-----------------|-----------------|----------------|----------------------------|
| Flower | $3.29 \pm .02$ | $3.01 \pm .06$ | $3.56 \pm .03$ | $3.62 \pm .02$ |
| CUB | $3.92 \pm .06$ | $3.52 \pm .08$ | $4.20 \pm .04$ | $4.31 \pm .06$ |
| COCO | $10.63 \pm .10$ | $6.36 \pm .20$ | $12.63 \pm .15$ | $20.13 \pm 22$ |

## C.2 HIGH-QUALITY IMAGES OF FIGURE 2

Due to space constraint, we provide relative small-size images in Fig. 2. Below we show the corresponding images with larger sizes.

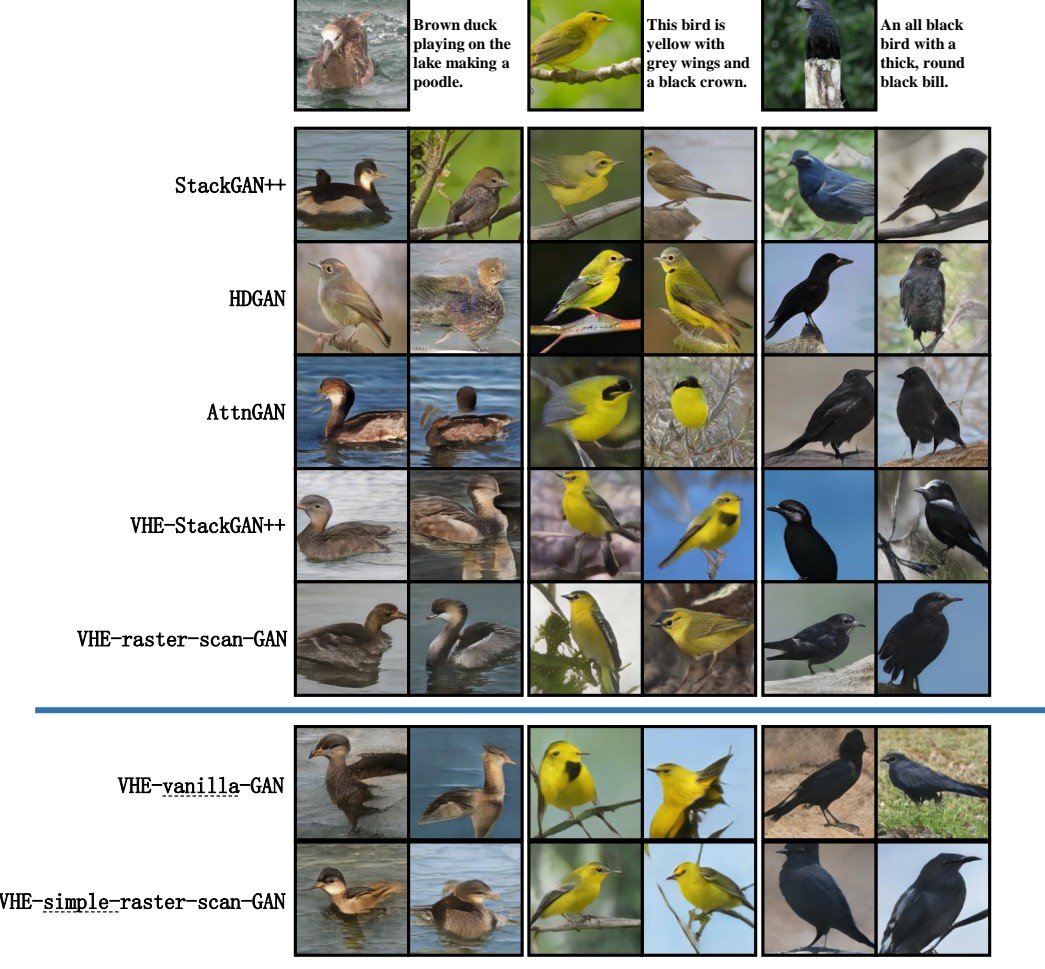

Figure 10: The images above the blue line are the larger-size replots of CUB Bird images in Figure 2, while the images below the blue line are results for ablation study.

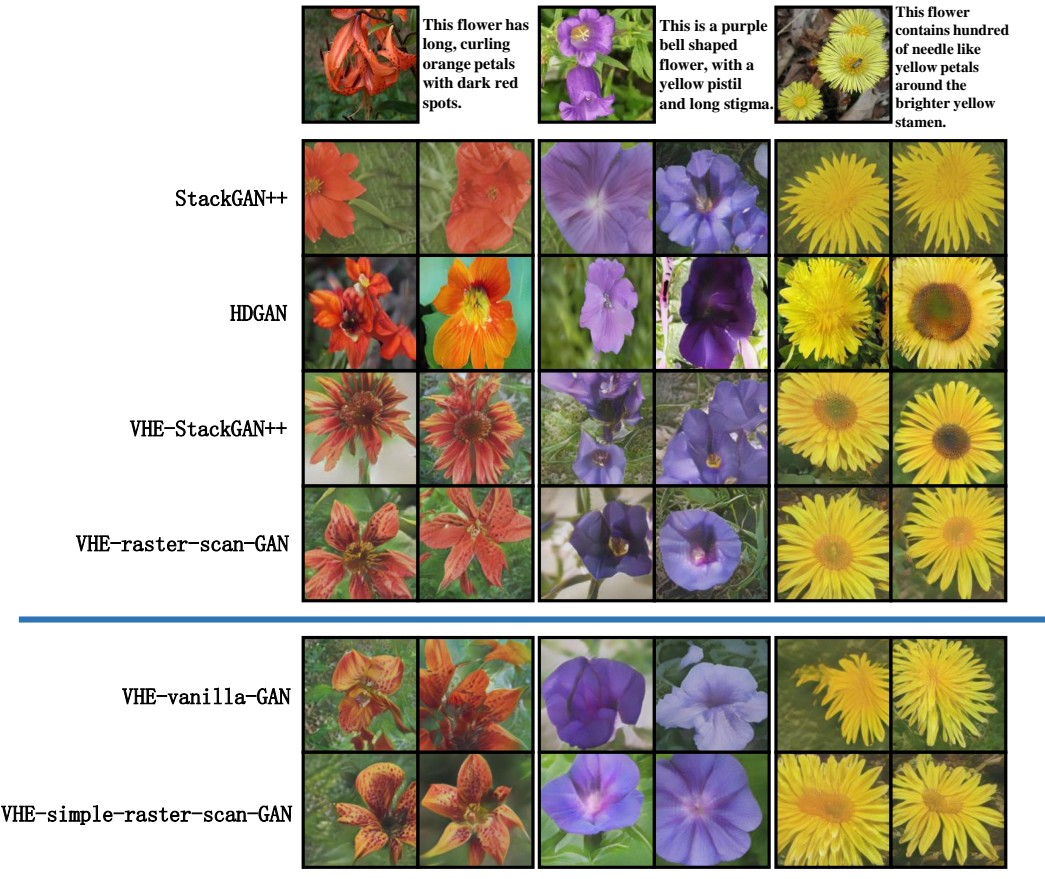

Figure 11: The images above the blue line are the larger-size replots of Flower images in Figure 2, while the images below the blue line are results for ablation study.

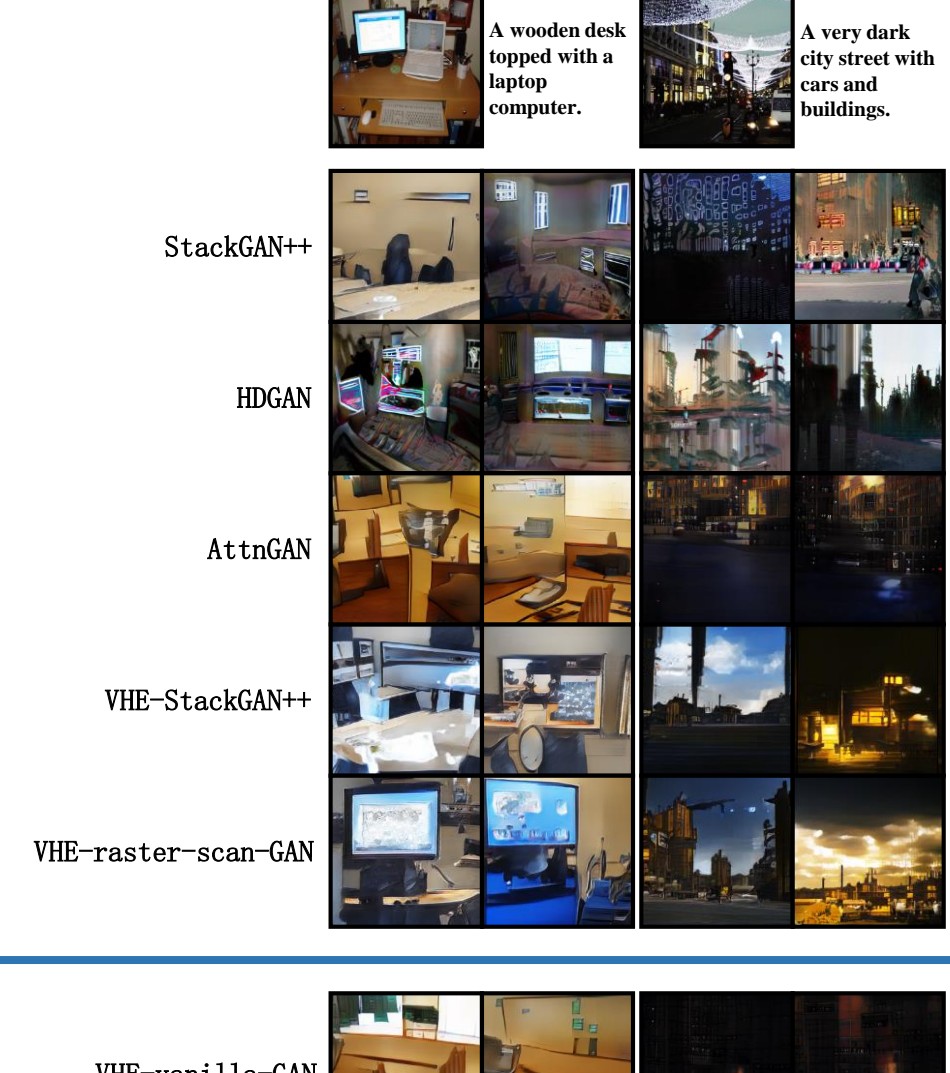

Figure 12: The images above the blue line are the larger-size replots of COCO images in Figure 2, while the images below the blue line are results for ablation study.

## C.3 MORE TEXT-TO-IMAGE GENERATION RESULTS ON COCO

COCO is a more challenging dataset than CUB and Flower, as it contains very diverse objects and scenes. We show in Fig. 13 more samples conditioned on different textural descriptions.

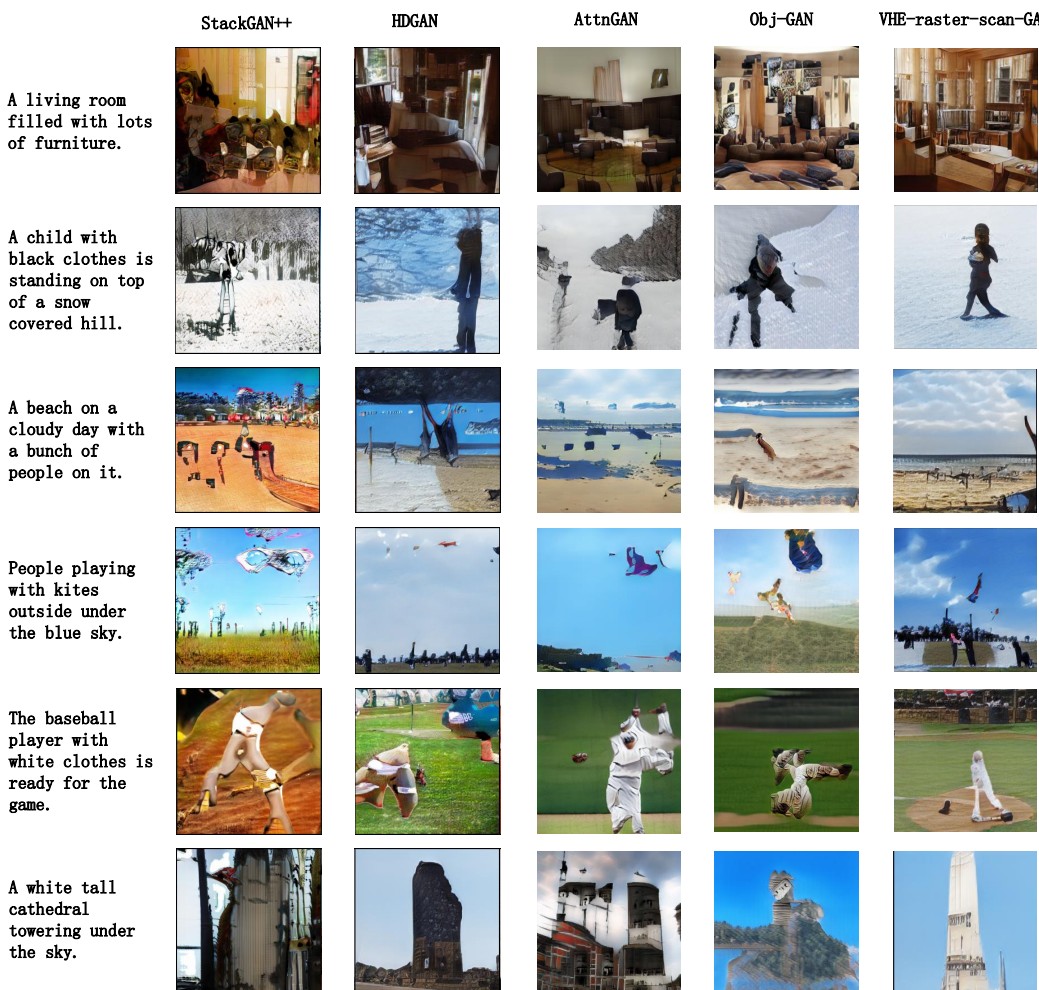

Figure 13: Example text-to-image generation results on COCO.

## C.4    LATENT SPACE INTERPOLATION

In addition to the latent space interpolation results of VHE-raster-scan-GAN in Fig. 3(c) of Section 3.1, below we provide more fine-gridded latent space interpolation in Figs. 15-18.

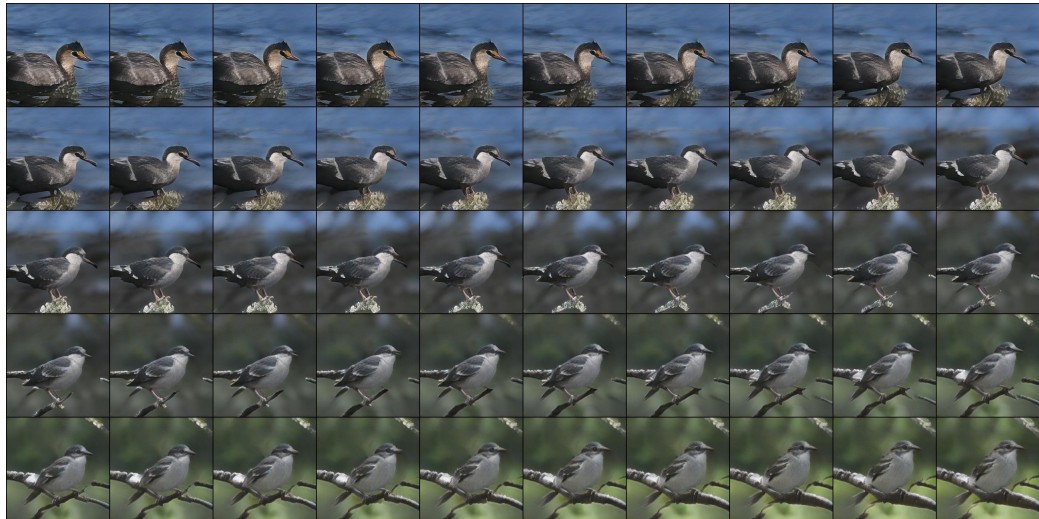

Figure 14: Example of latent space interpolation on CUB.

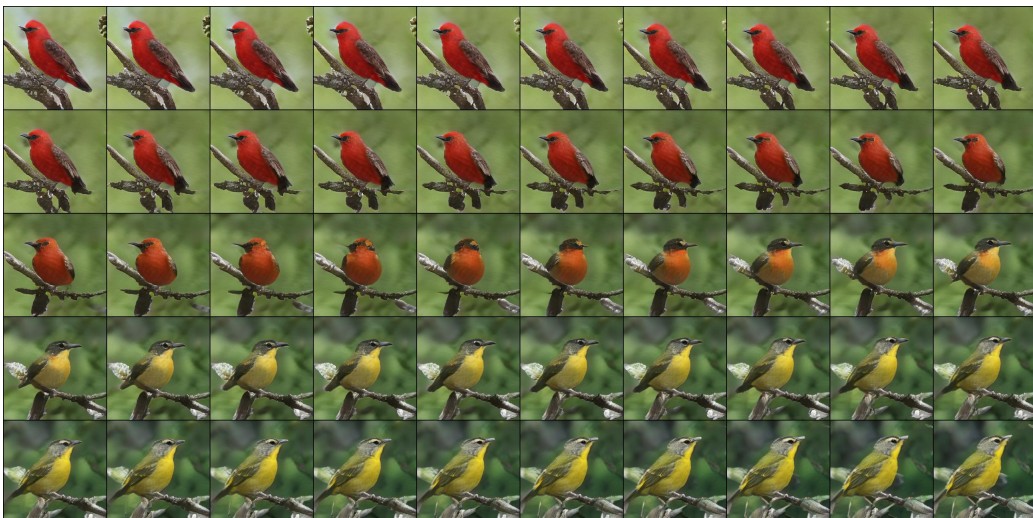

Figure 15: Example of latent space interpolation on CUB.

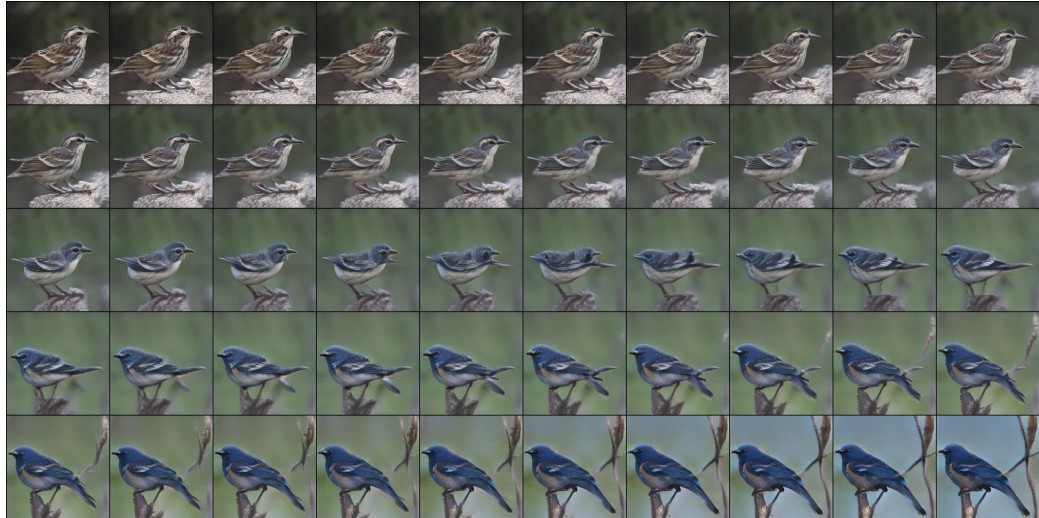

Figure 16: Example of latent space interpolation on CUB.

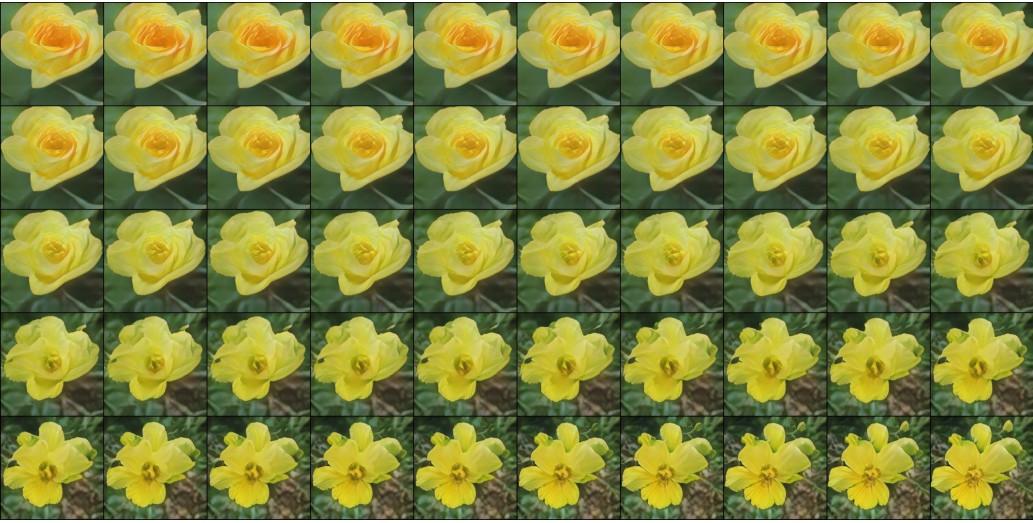

Figure 17: Example of latent space interpolation on Flower.

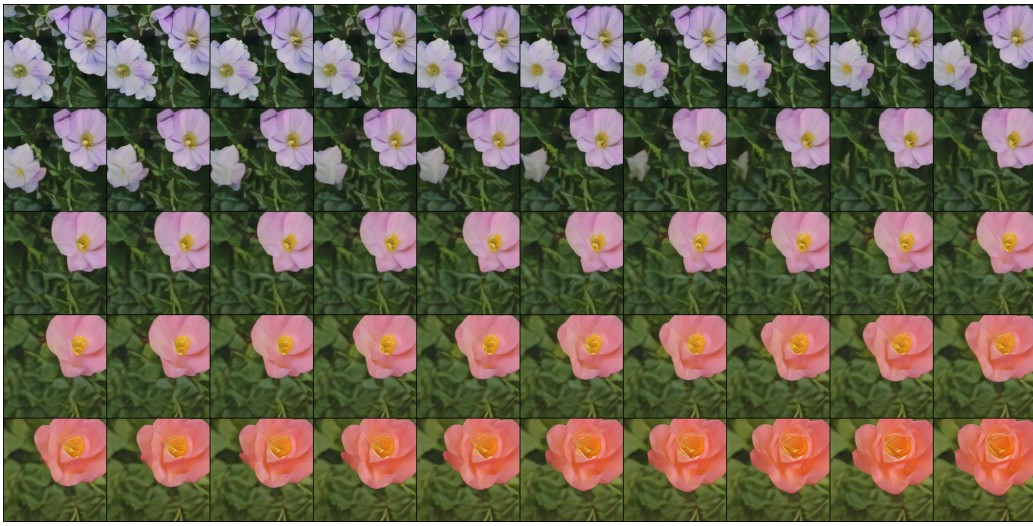

Figure 18: Example of latent space interpolation on Flower.

## C.5 IMAGE RETRIEVAL GIVEN A TEXT QUERY

For image $x_n$, we draw its BoW textual description $\hat{t}_n$ as $\hat{t}_n \,|\, \theta_n \sim p(t \,|\, \Phi, \theta_n)$, $\theta_n \,|\, x_n \sim q_{\Omega}(\theta \,|\, \Phi, x_n)$. Given the BoW textual description $t$ as a text query, we retrieve the top five images ranked by the cosine distances between $t$ and $\hat{t}_n$'s. Shown in Fig. 19 are three example image retrieval results, which suggest that the retrieved images are semantically related to their text queries in colors, shapes, and locations.

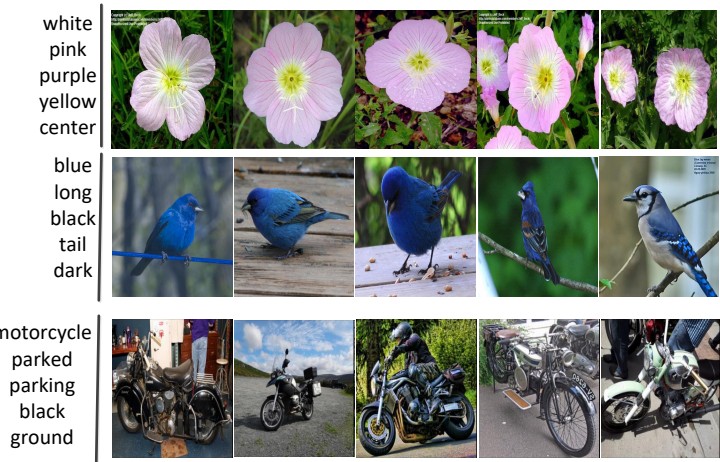

Figure 19: Top-5 retrieved images given a text query. Rows 1 to 3 are for Flower, CUB, and COCO, respectively.

## C.6 IMAGE REGENERATION

We note for VHE-GAN, its image encoder and GAN component together can also be viewed as an "autoencoding" GAN for images. More specifically, given image $x$, VHE-GAN can provide random regenerations using $G\left(q_{\Omega}\left(\theta \,|\, \Phi, x\right)\right)$. We show example image regeneration results by both VHE-StackGAN++ and VHE-raster-scan-GAN in Fig. 20. These example results suggest that the regenerated random images by the proposed VHE-GANs more of less resemble the original real image fed into the VHE image encoder.

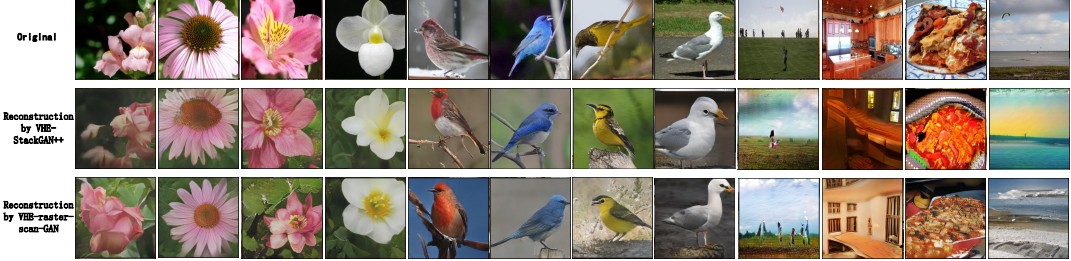

Figure 20: Example results of image regeneration using VHE-StackGAN++ and VHE-raster-scan-GAN. An original image is fed into the VHE image encoder, whose latent representation is then fed into the GAN image generator to generate a corresponding random image. The models in columns 1-4 are trained on Flower, columns 5-8 on CUB, and columns 9-12 on COCO.

## C.7 LEARNED HIERARCHICAL TOPICS IN VHE

The inferred topics at different layers and the inferred sparse connection weights between the topics of adjacent layers are found to be highly interpretable. In particular, we can understand the meaning of each topic by projecting it back to the original data space via $\left[\prod_{t=1}^{l-1} \Phi^{(t)}\right] \phi_k^{(l)}$ and understand the relationship between the topics by arranging them into a directed acyclic graph (DAG) and choose

its subnets to visualize. We show in Figs. 21, 22, and 23 example subnets taken from the DAGs inferred by the three-layer VHE-raster-scan-GAN of size 256-128-64 on Flower, CUB, and COCO, respectively. The semantic meaning of each topic and the connection weights between the topics of adjacent layers are highly interpretable. For example, in Figs. 21, the topics describe very specific flower characteristics, such as special colors, textures, shapes, and parts, at the bottom layer, and become increasingly more general when moving upwards.

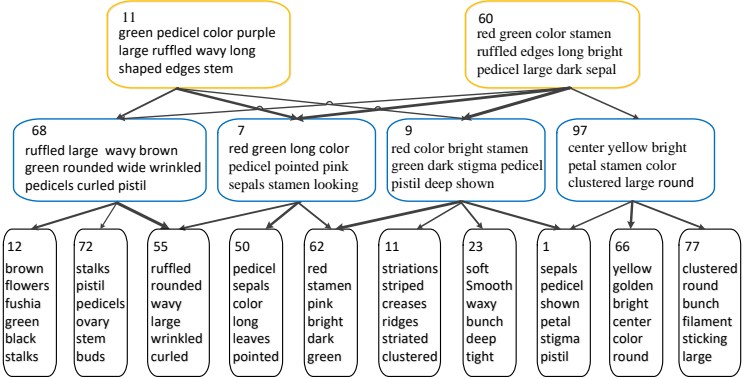

Figure 21: An example topic hierarchy taken from the directed acyclic graph learned by a three-layer VHE-raster-scan-GAN of size 256-128-64 on Flower.

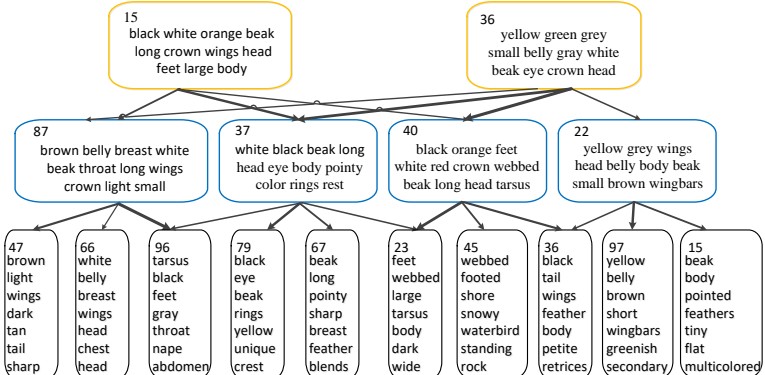

Figure 22: Analogous plot to Fig. 21 on CUB.

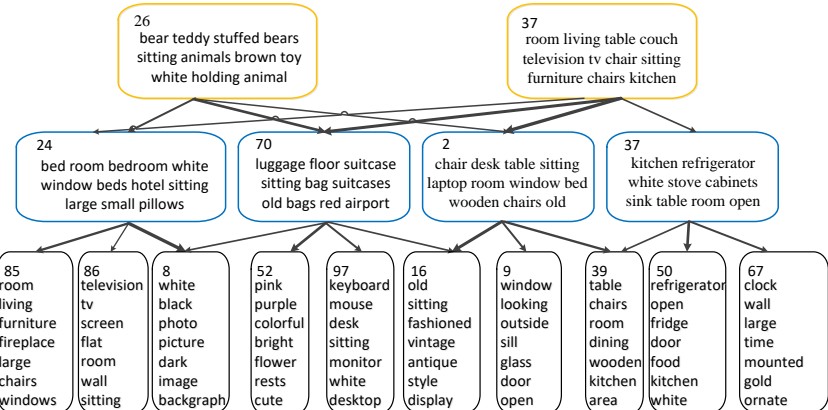

Figure 23: Analogous plot to Fig. 21 on COCO.

# D  SPECIFIC MODEL STRUCTURE IN VHE-STACKGAN++ AND VHE-RASTER-SCAN-GAN

## D.1  MODEL STRUCTURE OF VHE

In Fig. 24, we give the structure of VHE used in VHE-StackGAN++ and VHE-raster-scan-GAN, where $f(\boldsymbol{x})$ is the image features extracted by Inception v3 network and $\boldsymbol{\varepsilon}^{(l)} \sim \prod_{k=1}^{K_l} \text{Uniform}(\varepsilon_k^{(l)}; 0, 1)$. With the definition of $\boldsymbol{g}^{(0)} = f(\boldsymbol{x})$, we have

$$\boldsymbol{k}^{(l)} = \exp(\mathbf{W}_1^{(l)}\boldsymbol{g}^{(l)} + \boldsymbol{b}_1^{(l)}), \tag{12}$$

$$\boldsymbol{\lambda}^{(l)} = \exp(\mathbf{W}_2^{(l)}\boldsymbol{g}^{(l)} + \boldsymbol{b}_2^{(l)}), \tag{13}$$

$$\boldsymbol{g}^{(l)} = \ln[1 + \exp(\mathbf{W}_3^{(l)}\boldsymbol{g}^{(l-1)} + \boldsymbol{b}_3^{(l)})], \tag{14}$$

where $\mathbf{W}_1^{(l)} \in \mathbb{R}^{K_l \times K_l}$, $\mathbf{W}_2^{(l)} \in \mathbb{R}^{K_l \times K_l}$, $\mathbf{W}_3^{(l)} \in \mathbb{R}^{K_l \times K_{l-1}}$, $\boldsymbol{b}_1^{(l)} \in \mathbb{R}^{K_l}$, $\boldsymbol{b}_2^{(l)} \in \mathbb{R}^{K_l}$, and $\boldsymbol{b}_3^{(l)} \in \mathbb{R}^{K_l}$.

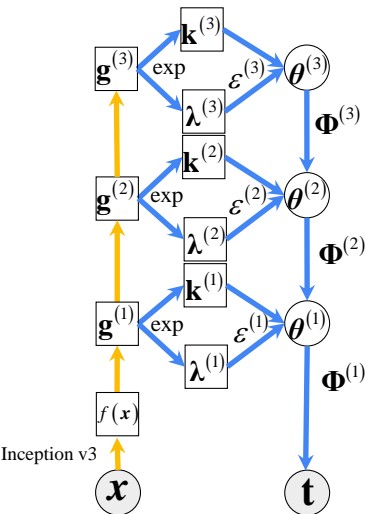

Figure 24: The architecture of VHE in VHE-StackGAN++ and VHE-raster-scan-GAN.

## D.2  MODEL OF VHE-STACKGAN++

In Section 2.2, we first introduce the VHE-StackGAN++, where the multi-layer textual representation $\{\boldsymbol{\theta}^{(1)}, \boldsymbol{\theta}^{(2)}, \cdots, \boldsymbol{\theta}^{(L)}\}$ is concatenated as $\boldsymbol{\theta} = \left[\boldsymbol{\theta}^{(1)}, \cdots, \boldsymbol{\theta}^{(L)}\right]$ and then fed into StackGAN++ (Zhang et al., 2017b). In Figs. 1 (a) and (b), we provide the model structure of VHE-StackGAN++. We also provide a detailed plot of the structure of StackGAN++ used in VHE-StackGAN++ in Fig. 25, where JCU is a specific type of discriminator; see Zhang et al. (2017b) for more details.

The same with VHE-raster-scan-GAN, VHE-StackGAN++ is also able to jointly optimize all components by merging the expectation in VHE and GAN to define its loss function as

$$\min_{\boldsymbol{\Omega}, \{G_i\}_{i=1}^3} \max_{\{D_i\}_{i=1}^3} \mathbb{E}_{p_{\text{data}}(\boldsymbol{x}_n, \boldsymbol{t}_n)} \mathbb{E}_{\prod_{l=1}^L q(\boldsymbol{\theta}_n^{(l)} \mid \boldsymbol{x}_n, \boldsymbol{\Phi}^{(1+1)}, \boldsymbol{\theta}_n^{(l+1)})} \big\{ -\log p(\boldsymbol{t}_n \mid \boldsymbol{\Phi}^{(1)}, \boldsymbol{\theta}_n^{(1)})$$
$$+ \sum_{l=1}^L \text{KL}[q(\boldsymbol{\theta}_n^{(l)} \mid \boldsymbol{x}_n, \boldsymbol{\Phi}^{(1+1)}, \boldsymbol{\theta}_n^{(l+1)}) \,||\, p(\boldsymbol{\theta}_n^{(l)} \mid \boldsymbol{\Phi}^{(1+1)}, \boldsymbol{\theta}_n^{(l+1)})]$$
$$+ \sum_{i=1}^3 [\log D_i(\boldsymbol{x}_{n,i}, \boldsymbol{\theta}_n) + \log(1 - D_i(G_i(\boldsymbol{\theta}_n), \boldsymbol{\theta}_n))]\big\}. \tag{15}$$

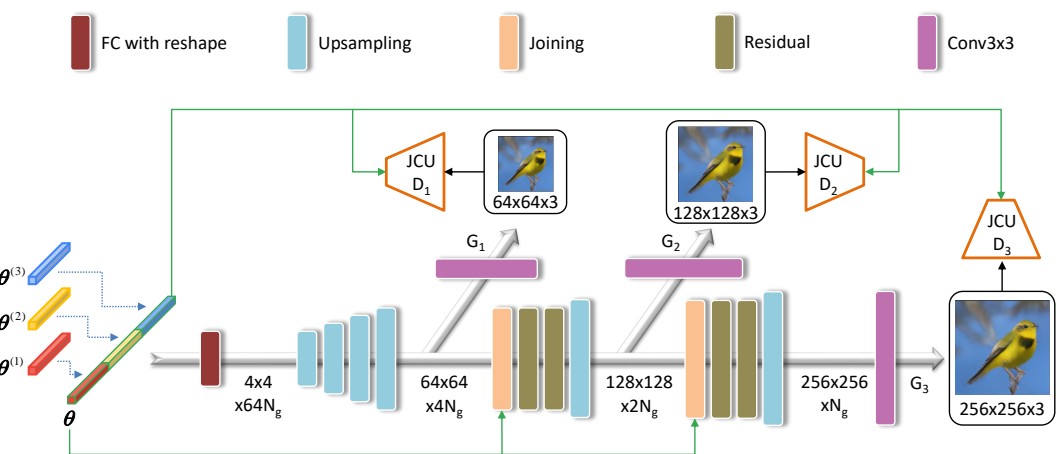

Figure 25: The structure of Stack-GAN++ in VHE-StackGAN++, where JCU is a type of discriminator proposed in Zhang et al. (2017b).

## D.3 STRUCTURE OF RASTER-SCAN-GAN

In Fig. 26, we provide a detailed plot of the structure of the proposed raster-scan-GAN.

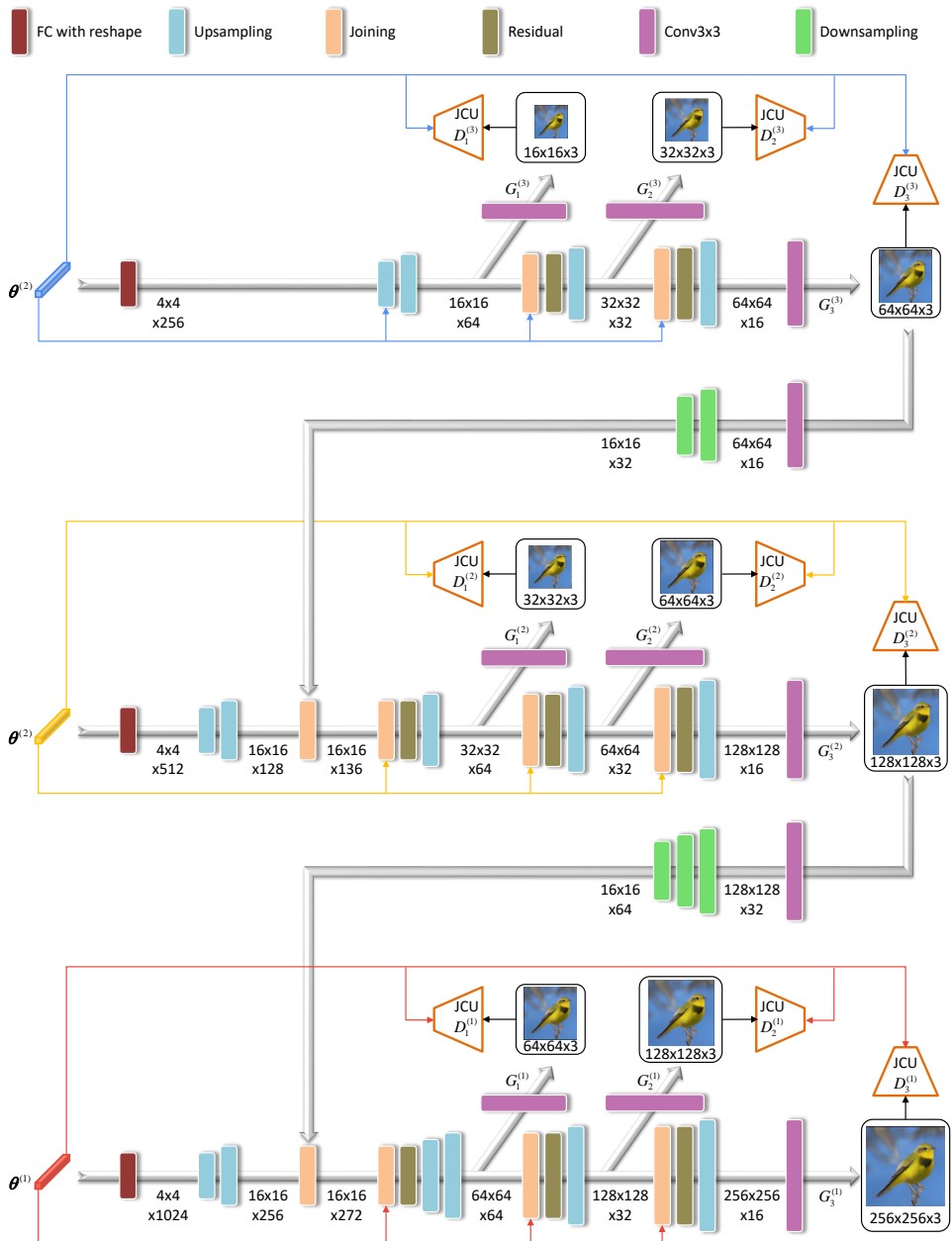

Figure 26: The structure of raster-scan-GAN in VHE-raster-scan-GAN, where JCU is a type of discriminator proposed in Zhang et al. (2017b).

## E  Joint optimization for VHE-raster-scan-GAN

Based on the loss function of VHE-raster-scan-GAN (9), with TLASGR-MCMC (Cong et al., 2017) and WHAI (Zhang et al., 2018a), we describe in Algorithm 1 how to perform mini-batch based joint update of all model parameters.

---

**Algorithm 1** Hybrid TLASGR-MCMC/VHE inference algorithm for VHE-raster-scan-GAN.

---

Initialize encoder parameters $\mathbf{\Omega}$, topic parameters of PGBN $\{\mathbf{\Phi}^{(l)}\}_{1,L}$, generator $G$, and discriminator $D$.

**for** $iter = 1, 2, \cdots$ **do**

Randomly select a mini-batch containing $N$ image-text pairs $\boldsymbol{d} = \{\boldsymbol{x}_n, \boldsymbol{t}_n\}_{n=1}^N$;

Draw random noise $\left\{\varepsilon_n^{(l)}\right\}_{n=1,l=1}^{N,L}$ from uniform distribution;

Calculate $\nabla_D \mathcal{L}(D, G, \mathbf{\Omega} \,|\, \boldsymbol{x})$;

Calculate $\nabla_G \mathcal{L}(D, G, \mathbf{\Omega} \,|\, \boldsymbol{x})$;

Calculate $\nabla_{\mathbf{\Omega}} L$ by the aid of $\left\{\varepsilon_n^{(l)}\right\}_{n=1,l=1}^{N,L}$;

Update $D$ as $D = D + \nabla_D \mathcal{L}(D, G, \mathbf{\Omega} \,|\, \boldsymbol{x})$;

Update $G$ as $G = G - \nabla_G \mathcal{L}(D, G, \mathbf{\Omega} \,|\, \boldsymbol{x})$;

Update $\mathbf{\Omega}$ as $\mathbf{\Omega} = \mathbf{\Omega} - \nabla_{\mathbf{\Omega}} L$;

Sample $\{\boldsymbol{\theta}_n^{(l)}\}_{l=1}^L$ from (6) given $\mathbf{\Omega}$ and $\{\mathbf{\Phi}^{(l)}\}_{l=1}^L$, and use $\{\boldsymbol{t}\}_{n=1}^N$ to update topics $\{\mathbf{\Phi}^{(l)}\}_{l=1}^L$ according to TLASGR-MCMC;

**end for**

---

## F  Data description on CUB, Flower, and COCO with training details

In image-text multi-modality learning, CUB (Wah et al., 2011), Flower (Nilsback & Zisserman, 2008) and COCO (Lin et al., 2014) are widely used datasets.

**CUB** (`http://www.vision.caltech.edu/visipedia/CUB-200-2011.html`): CUB contains 200 bird species with 11,788 images. Since 80% of birds in this dataset have object-image size ratios of less than 0.5 (Wah et al., 2011), as a preprocessing step, we crop all images to ensure that bounding boxes of birds have greater-than-0.75 object-image size ratios, which is the same with all related work. For textual description, Wah et al. (2011) provide ten sentences for each image and we collect them together to form BoW vectors. Besides, for each species, Elhoseiny et al. (2017a) provide its encyclopedia document for text-based ZSL, which is also used in our text-based ZSL experiments.

For CUB, there are two split settings: the hard one and the easy one. The hard one ensures that the bird subspecies belonging to the same super-category should belong to either the training split or test one without overlapping, referred to as CUB-hard (CUB-H in our manuscript). A recently used split setting (Qiao et al., 2016; Akata et al., 2015) is super-category split, where for each super-category, except for one subspecies that is left as unseen, all the other are used for training, referred to as CUB-easy (CUB-E in our manuscript). For CUB-H, there are 150 species containing 9410 samples for training and 50 species containing 2378 samples for testing. For CUB-E, there are 150 species containing 8855 samples for training and 50 species containing 2933 samples to testing. We use both of them the for the text-based ZSL, and only CUB-E for all the other experiments as usual.

**Flower** `http://www.robots.ox.ac.uk/~vgg/data/flowers/102/index.html`: Oxford-102, commonly referred to as Flower, contains 8,189 images of flowers from 102 different categories. For textual description, Nilsback & Zisserman (2008) provide ten sentences for each image and we collect them together to form BoW vectors. Besides, for each species, Elhoseiny et al. (2017a) provide its encyclopedia document for text-based ZSL, which is also used in our text-based ZSL experiments in section 4.2.2. There are 82 species containing 7034 samples for training and 20 species containing 1155 samples for testing.

For text-based ZSL, we follow the same way in Elhoseiny et al. (2017a) to split the data. Specifically, five random splits are performed, in each of which $4/5$ of the classes are considered as "seen classes" for training and $1/5$ of the classes as "unseen classes" for testing. For other experiments, we follow Zhang et al. (2017b) to split the data.

**COCO** `http://cocodataset.org/#download`: Compared with Flower and CUB, COCO is a more challenging dataset, since it contains images with multiple objects and diverse backgrounds. To show the generalization capability of the proposed VHE-GANs, we also utilize COCO for evaluation. Following the standard experimental setup for COCO (Reed et al., 2016; Zhang et al., 2017b), we directly use the pre-split training and test sets to train and evaluate our proposed models. There are 82081 samples for training and 40137 samples for testing.

**Training details:** we train VHE-rater-scan-GAN in four Nvidia GeForce RTX2080 TI GPUs. The experiments are performed with mini-batch size 32 and about 30.2G GPU memory space. We run 600 epochs to train the models on CUB and Flower, taking about 797 seconds for CUB-E and 713 seconds for Flower for each epoch. We run 100 epochs to train the models on COCO, taking about 6315 seconds for each epoch. We use the Adam optimizer (Kingma & Ba, 2014) with learning rate $2 \times 10^{-4}$, $\beta_1 = 0.5$, and $\beta_2 = 0.999$ to optimize the parameters of the GAN generator and discriminator, and use Adam with learning rate $10^{-4}$, $\beta_1 = 0.9$, and $\beta_2 = 0.999$ to optimize the VHE parameters. The hyper-parameters to update the topics $\mathbf{\Phi}$ with TLASGR-MCMC are the same with those in Cong et al. (2017).

## G  ADDITIONAL DISCUSSION ON OBJ-GAN

Focusing on the COCO dataset, the recently proposed Obj-GAN (Li et al., 2019) exploits more side information, including the bounding boxes and labels of objects existing in the images, to perform text-to-image generation. More specifically, Obj-GAN first trains an attentive sequence to sequence model to infer the bounding boxes given a text $\boldsymbol{t}$:

$$B_{1:T} = [B_1, B_2, \cdots, B_T] = G_{\text{box}}(\boldsymbol{e}), \tag{16}$$

where, $\boldsymbol{e}$ are the pre-trained bi-LSTM word vectors of $\boldsymbol{t}$, $B_t = (l_t, b_t)$ consists of the class label of the $t$th object and its bounding box $b = (x, y, w, h) \in \mathbb{R}^4$. Given the bounding boxes $B_{1:T}$, Obj-GAN learn a shape generator to predict the shape of each object in its bounding box:

$$\hat{M}_{1:T} = G_{\text{shape}}(B_{1:T}, \boldsymbol{z}_{1:T}), \tag{17}$$

where $\boldsymbol{z}_t \sim \mathcal{N}(0, 1)$ is a random noise vector. Having obtained $B_{1:T}$ and $\hat{M}_{1:T}$, Obj-GAN trains an attentive multi-stage image generator to generate the images conditioned on $B_{1:T}$, $\hat{M}_{1:T}$, and $\boldsymbol{e}$.

Although Obj-GAN achieves a better FID on COCO, it has two major limitations in practice. First, it is not always possible to obtain accurate bounding boxes and labels of objects in the image; even they can be acquired by manual labeling, it is often time and labor consuming, especially on large datasets. Second, each word is associated with one fixed bounding box; in other words, given one sentence, the locations of the objects in the generated images are fixed, which clearly hurts the diversity of the Obj-GAN generated images, as shown in Fig. 27.

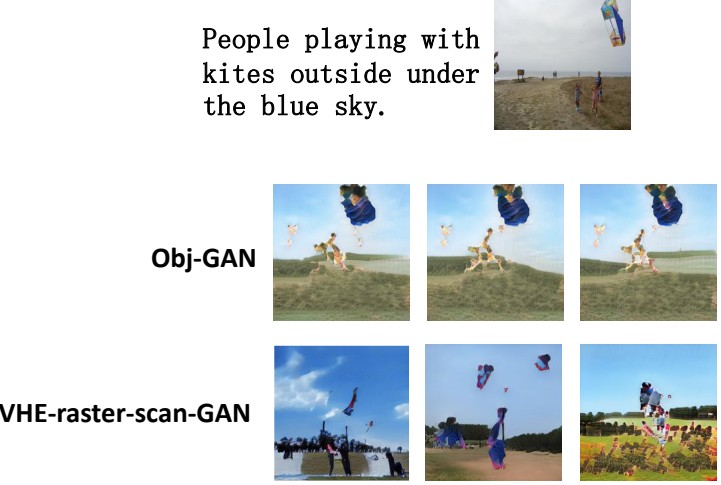

Figure 27: The generated random images of Obj-GAN given text lack diversity.

