# OpenReview forum: "Variational Hetero-Encoder Randomized GANs for Joint Image-Text Modeling"
_ICLR.cc/2020/Conference — Accept (Poster)_

### Official Review · AnonReviewer4 · 2019-10-23
**Official Blind Review #2**

**Rating:** 3

**Review:**

This paper proposed VHE-GAN for the text-to-image generation task. The proposed method utilizes the off-the-shell modules and feeds the VHE variational posterior into the generator. The experiments are conducted on three datasets.

The motivation for the paper is not clear. Most of the components used, such as text-encoder, image-encoder, generator-discriminator follow previous works. Therefore, the authors should claim how the proposed VHE variational posterior can help the task. However, I did not see the clear motivation for this part.

Besides the basic version VHE-StackGAN++, it proposed another version  VHE-raster-scan-GAN. However, the paper also fails to tell the intuition of the deep topic model and PGBN text decoder.

The experimental results are not solid. The comparison only included old baselines. However, several recent state-of-the-art approaches are missing: a. attnGAN (CVPR18), b. TA-GAN (NIPS18), c. Object-GAN (CVPR19). Without these comparisons, it is difficult to evaluate how the method works. In addition, the paper does not provide an ablation study to analyze the effect of each component proposed (e.g., Poisson gamma belief network, a deep topic mode).

**Experience Assessment:**

I have published one or two papers in this area.

**Review Assessment: Checking Correctness Of Derivations And Theory:**

N/A

**Review Assessment: Checking Correctness Of Experiments:**

I carefully checked the experiments.

**Review Assessment: Thoroughness In Paper Reading:**

I read the paper thoroughly.

---

> ### Author Response · Authors · 2019-11-14
> **Additional response to AnonReviewer4**
>
> We thank AnonReviewer4 for the comments. We'd like to strongly argue against these comments that were used to support the low rating. Please see our point-by-point response below.
>
> First, we emphasize that the proposed VHE-GANs are deep generative models that can not only perform text-to-image generation tasks, but also many other tasks, such as text based zero shot learning (image-to-text) and noise to image-text pairs generation. Please see our "Response to All" on the versatility of the proposed models.
>
> Second, a key motivation of the paper is to build VHE-raster-scan-GAN. While it does include a variety existing modeling components, both the VHE-GAN framework, which integrates various components for bidirectional image-text modeling, and the raster-scan structure, which leads to state-of-the-art results in a variety of tasks, are first proposed in this paper. Please see our "Response to All" on additional ablation studies that have been added to better demonstrate the importance of having both the stacking structure and raster-scan structure.
>
> Third, we did explain the intuition of why using the the PGBN deep topic model (together with the raster-scan structure) in multiple places, such as in the last paragraph of Section 2.2, first paragraph of Section 2.3, and last paragraph of Section 3.1. More specifically, a key intuition behind VHE-raster-scan-GAN is to use the PGBN deep topic model to help capture hierarchical semantic structures, which are related to coarse-to-fine visual concepts with the help of the raster-scan structure, as visually demonstrated in Figs. 4, 5, 21-23 and a variety of comparisons between VHE-StackGAN++ and VHE-raster-scan-GAN. Higher layer topics are mainly related to the general shapes and colors of objects, or backgrounds, while the lower layer ones are focused on finer details. VHE-raster-scan-GAN exploits the hierarchical semantic structure, which matches coarse-to-fine visual concepts, to gradually refine its generation under the proposed VHE-GAN framework.
>
> Fourth, we note in the original submission, we did include AttnGAN for comparison in Table 1, and an ablation study in Table 4 to examine the effect of varying the depth of PGBN used in VHE-raster-scan-GAN, as discussed in the last paragraph of Section 3.2. In Fig. 2 of our original manuscript, we did not show the results of AttnGAN as it provided  no results on Flower.
>
> To address your concerns, we have now added AttnGAN into Fig. 2 and more results of AttnGAN in Appendices C.2 and C.3 in our revised paper, included two additional variations of VHE-raster-scan-GAN for ablation studies, and added both TA-GAN and Obj-GAN into comparisons. The newly added discussions and results have been highlighted in blue in the revised paper. Please see our "Response to All" for more details.

---

### Official Review · AnonReviewer3 · 2019-10-23
**Official Blind Review #3**

**Rating:** 8

**Review:**

This paper proposes a combined architecture for image-text modeling. Though the proposed architecture is extremely detailed, the authors explain clearly the overarching concepts and methods used, within limited space. The experimental results are extremely strong, especially on sub-domains where conditional generative models have historically struggled such as images with angular, global features - often mechanical or human constructed objects. "Computers" and "cars" images in Figure 2 show this quite clearly. The model also functions for tagging and annotating images - performing well compared to models designed *only* for this task.

The authors have done a commendable job adding detail, further analysis, and experiments in the appendix of the paper. Combined with the included code release, this paper should be of interest to many.

My chief criticisms come for the density of the paper - while it is difficult to dilute such a complex model to 8 pages, and the included appendix clarifies many questions in the text body, it would be worth further passes through the main paper with a specific focus on clarity and brevity, to aid in the accessibility of this work.

As usual, more experiments are always welcome, and given the strengths of GAN based generators for faces a text based facial image generator could have been a great addition. The existing experiments are more than sufficient for proof-of-concept though.

Finally, though this version of the paper includes code directly in a google drive link it would be ideal for the final version to reference a github code link - again to aid access to interested individuals. Being able to read the code online, without downloading and opening locally can be nice, along with other benefits from open source release. However the authors should release the code however they see fit, this is more of a personal preference on the part of this reviewer.

To improve my score, the primary changes would be more editing and re-writing, focused on clarity and brevity of the text in the core paper.

**Experience Assessment:**

I have published one or two papers in this area.

**Review Assessment: Checking Correctness Of Derivations And Theory:**

I assessed the sensibility of the derivations and theory.

**Review Assessment: Checking Correctness Of Experiments:**

I assessed the sensibility of the experiments.

**Review Assessment: Thoroughness In Paper Reading:**

I read the paper at least twice and used my best judgement in assessing the paper.

---

> ### Author Response · Authors · 2019-11-14
> **Additional response to AnonReviewer3**
>
> Thank you for your comments and suggestions. We have revised our paper to add additional comparisons to recent work, more ablation studies, and more experimental results, with the major additions highlighted in blue. Please see our "Response to All" for details. Below please find our additional response.
>
> To enhance clarity and brevity, we have modified our ablation studies to better focus on presenting the proposed VHE-raster-scan-GAN.
>
> We have added the task of text-to-face-image generation. More specifically, we have trained our models on the CelebA dataset, where each facial image is described by 40 textual attributes. We have added preliminary results of text-attributes-to-face-image generation to Appendix B of the revised paper. Note limited by the rebuttal time and our computational resource, these examples facial images at 128 *128 resolution were generated from a relatively small network trained with only 20 epochs. We are working on adding more training epochs and increasing the network size and image resolution to further improve these preliminary results, which will be included in our next revision. We are also seeking facial image datasets with textual descriptions beyond only attributes (it seems that we might need to build them by our own), which will be our future work.
>
> We released the code on Google drive to help better preserve anonymity. After the acceptance of this paper, we will release it in GitHub for better access.

---

### Official Review · AnonReviewer1 · 2019-10-24
**Official Blind Review #1**

**Rating:** 6

**Review:**

Summary: The authors design a new model for bidirectional joint image-text modeling using a variational hetero-encoder
(VHE) randomized generative adversarial network (GAN) that integrates a probabilistic text decoder, probabilistic image encoder, and GAN into an end-to-end multimodal model. Their proposed VHE-GAN model encodes an image to decode its associated text and feeds the variational posterior as the source of randomness into the GAN image generator. The authors also incorporate a deep topic model, a ladder-structured image encoder, and StackGAN++ into their framework for improved photo-realistic images.

Strengths:
- The authors have proposed a nice multimodal model that allows inference of latent variables given only text or image, and also allows realistic synthesis of images from images, text, or noise.
- The paper is quite dense but generally well written.

Weaknesses:
- The experimental comparison only included old baselines and the authors should compare to some more recent work such as TA-GAN (NIPS18), and Object-GAN (CVPR19).
- It would help if the paper contained more ablation studies across different modules that the framework uses.

### Post rebuttal ###
Thank you for your detailed answers to my questions.

**Experience Assessment:**

I have read many papers in this area.

**Review Assessment: Checking Correctness Of Derivations And Theory:**

I assessed the sensibility of the derivations and theory.

**Review Assessment: Checking Correctness Of Experiments:**

I assessed the sensibility of the experiments.

**Review Assessment: Thoroughness In Paper Reading:**

I read the paper at least twice and used my best judgement in assessing the paper.

---

> ### Author Response · Authors · 2019-11-14
> **Additional response to AnonReviewer1**
>
> Thank you for your comments and suggestions. Following your suggestions, we have performed comparisons against both TA-GAN and Obj-GAN, and included two additional variations of VHE-raster-scan-GAN for ablation studies. The newly added discussions and results have been highlighted in blue in the revised paper. Please see our "Response to All" for more details.

---

### Author Response · Authors · 2019-11-14
**Response to All**

We thank the reviewers for their valuable comments and suggestions that have helped us to improve the paper. In the revised paper, we have performed comparisons to both TA-GAN (NeurIPS2018) and Obj-GAN (CVPR2019), introduced additional ablation studies, and added results on text-to-face-image generation. The main additions to the paper are highlighted in blue. Below we first respond to all reviewers.

1) Additional baselines are included:

In the original version, for the text-to-image generation task, we focused on comparing to StackGAN++, HDGAN, and AttnGAN, as shown in Table 1, mainly because these models and the proposed ones only need texts as the input to generate images at the testing stage. In the revised paper, we have followed your suggestions to add comparisons to TA-GAN and Obj-GAN, which both require additional information beyond texts as the input.

1.1) Comparison with TA-GAN:

TA-GAN is not applicable to the conventional text-to-image (T2I) generation task as it also requires a real image as part of the input. For this reason, we have now compared to it on the same image-to-text and text-to-image retrieval tasks considered in TA-GAN. We have added Section 3.3 to provide quantitative evaluations, which show that VHE-raster-scan-GAN performs slightly better than TA-GAN on image-to-text retrieval while slightly worse on text-to-image retrieval. Note TA-GAN needs to extract its text features based on the fastText model  (Bojanowski et al., 2017) pre-trained on a large corpus, while VHE-raster-scan-GAN learns everything directly from the current dataset in an end-to-end manner.

1.2) Comparison with Obj-GAN:

We have added Obj-GAN for comparison into Table 1 and example images in Figs. 13 and 27, with some discussion added in Section 3.1 and Appendix G. For T2I generation, while Obj-GAN, with the help of object bounding boxes and labels, clearly outperforms the other methods in terms of the FID score, it underperforms VHE-raster-scan-GAN in terms of both the IS score and the visual quality and diversity of the generated images.  Besides the need of a pre-trained bi-LSTM word vectors, similar to StackGAN++, HDGAN, and AttnGAN, it also needs more side-information: the bounding boxes and labels of the objects in the images to train box and shape generators. By contrast, our proposed models build relationships between the original images and texts directly, without any extra pre-trained linguistic models or side-information on object bounding boxes and labels, making them easier to use in practice.

2) Additional ablation studies are performed:

We have added additional ablation studies across different modules. The original version included the following models for ablation studies: PGBN+StackGAN++, VHE-StackGAN++, VHE-L3 (without GAN part), and VHE-raster-scan-GAN with different layers of document decoder. In the revised paper, as shown in Fig. 1, we have added two new models for ablation study, whose results are shown in Table 2, and added related discussion in Section 3.1. These additional ablation studies have further demonstrated the benefits of having both the stacking structure and raster-scan structure that is invented in this paper.

3) On the versatility of the proposed deep generative models:

We'd like to emphasize that our proposed deep generative models achieve state-of-the-art results in T2I tasks, while not being designed solely for that purpose. They provide bidirectional transformations that make them distinct from existing models that can only provide unidirectional transformations. As discussed in the paper, ours are able to perform not only T2I tasks given short texts, but also T2I tasks given textual attributes (Fig. 3a and Fig. 7) or long documents (Fig. 3b and Fig. 8), image to text generation (Fig. 6a), random noise to image and text pairs generation (Fig. 6b), and image to image reconstruction (Fig. 19). Moreover, VHE-raster-scan-GAN exploits the explicit hierarchical relationships between images and text (Figs. 4 and 5) to help us to achieve the state-of-the-art results on text-based zero-short-learning tasks (Table 4).

Below please find our response to each individual reviewer.

---

### Decision · Program_Chairs · 2019-12-19

**Decision:**

Accept (Poster)

**Comment:**

This paper proposes a bidirectional joint image-text model using a variational hetero-encoder (VHE) randomized generative adversarial network (GAN). The proposed VHE-GAN model encodes an image to decode its associated text. Three reviewers have split reviews. Reviewer #3 is overall positive about this work. Reviewer #1 rated weak acceptance, while request more comparison with latest works. Reviewer  #2 rated weak reject raised concerns on the motivation of the approach, the lack of ablation and lack of comparison with the latest work. During the rebuttal, the authors provide additional comparison and ablation, which seem to address the major concerns. Given the overall positive feedback and the quality of rebuttal, the AC recommends acceptance.